# Non-linear response of summertime marine productivity to increased meltwater discharge around Greenland

M.J. Hopwood [1], D. Carroll[2], T.J. Browning [1], L. Meire [3,4], J. Mortensen [4], S. Krisch[1] & E.P. Achterberg[1]

Runoff from the Greenland Ice Sheet (GrIS) is thought to enhance marine productivity by adding bioessential iron and silicic acid to coastal waters. However, experimental data suggest nitrate is the main summertime growth-limiting resource in regions affected by meltwater around Greenland. While meltwater contains low nitrate concentrations, subglacial discharge plumes from marine-terminating glaciers entrain large quantities of nitrate from deep seawater. Here, we characterize the nitrate fluxes that arise from entrainment of seawater within these plumes using a subglacial discharge plume model. The upwelled flux from 12 marine-terminating glaciers is estimated to be >1000% of the total nitrate flux from GrIS discharge. This plume upwelling effect is highly sensitive to the glacier grounding line depth. For a majority of Greenland's marine-terminating glaciers nitrate fluxes will diminish as they retreat. This decline occurs even if discharge volume increases, resulting in a negative impact on nitrate availability and thus summertime marine productivity.

[1] Marine Biogeochemistry, GEOMAR Helmholtz Centre for Ocean Research, Kiel 24148, Germany. [2] Jet Propulsion Laboratory, California Institute of Technology, Pasadena, CA 91109, USA. [3] Royal Netherlands Institute for Sea Research, and Utrecht University, Korringaweg 7, 4401 NT Yerseke, The Netherlands. [4] Greenland Climate Research Centre, Greenland Institute of Natural Resources, PO BOX 570, 3900 Nuuk, Greenland. Correspondence and requests for materials should be addressed to M.J.H. (email: mhopwood@geomar.de)

A sustained increase in the annual volume of liquid and solid ice discharge from the Greenland Ice Sheet (GrIS) has been observed in recent decades, with the mean annual mass loss of 186 Gt between 2003–2010 being more than double that from 1983–2003[1,2]. The seasonal discharge pulse has a range of physical and biogeochemical impacts on receiving waters as it can affect nutrient supply[3,4], carbonate chemistry[5,6], fjord-scale circulation[7,8], and the seasonal pattern of primary production[9,10]. It has been widely hypothesized that increasing discharge from the GrIS will positively affect marine primary productivity either by iron (Fe)[4,11] or macronutrient fertilization[3,12]. However, the immediate fertilization potential of meltwater-derived nutrients will depend on the identity of the resource(s) that limit(s) summertime marine productivity around Greenland and how fluxes of this resource scale with GrIS discharge volume.

Nutrient fluxes from the GrIS to the coastal ocean are presently assumed to increase proportionately with discharge volume[3,13,14]. However, this assumption fails to consider both the contrasting mechanisms of nutrient delivery for land- and marine-terminating glaciers[15], and the potential for the low macronutrient concentration in meltwater to dilute the macronutrient content of seawater[16]. While runoff from land-terminating glaciers is assumed to be a nutrient source to the marine environment[3,4,12], it makes an almost negligible contribution to the $NO_3$ available for Arctic or North Atlantic productivity. Furthermore, runoff from land-terminating glaciers can suppress marine productivity through stratifying coastal waters, which impedes vertical macronutrient supply[17]. Consequently, the net effect of increasing runoff from glaciers on macronutrient availability in the marine environment can be negative[15].

Fluxes of Fe to the coastal ocean are sustained from both land- and marine-terminating glaciers[18,19]. It is well demonstrated that significant (~90–99%) losses of glacially sourced dissolved Fe occur upon mixing with seawater due to flocculation, which diminishes the flux of this micronutrient[19–21]. Less well understood are the physical mixing processes induced by subglacial discharge plumes, which may also lead to a complex non-linear relationship between meltwater discharge volume and the magnitude of the induced nutrient fluxes from upwelling[22]. The upwelling of macronutrient-rich bottom waters entrained within subglacial discharge plumes has recently been shown to constitute the dominant source of $NO_3$ supplied to the photic zone downstream of two Greenlandic marine-terminating glaciers[15,23] and this upwelling mechanism may be responsible for maintaining unusual patterns of seasonal primary production in these systems. The seasonal cycle of primary production across the high-latitude North Atlantic, including the Labrador Sea and Irminger Basin, is generally characterized by a pronounced spring bloom commencing in April or May, followed by a less productive summer period[24,25]. Few studies have investigated the seasonal development of primary production close to Arctic marine-terminating glaciers, yet observations from Godthåbsfjord (SW Greenland) suggest that discharge from the GrIS is associated with a pronounced summer bloom in July or August, which accounts for approximately half of annual primary production[9,15,26].

While upwelled fluxes of macronutrients are potentially important for driving summertime productivity downstream of marine-terminating glaciers[15], it remains unknown how this induced macronutrient supply responds to changes in subglacial discharge volume and the bathymetric-controlled position of a marine-terminating glacier's grounding line in the water column. Because upwelled macronutrient fluxes may be several orders of magnitude greater than any fluxes arising directly from meltwater itself, this represents the single greatest uncertainty in how macronutrient fluxes to surface waters around the GrIS will respond to the combined effects of increasing discharge and glacier retreat.

In order to understand the potential influence of increased discharge from the GrIS on summer marine productivity, here we investigate how macronutrient availability changes during the meltwater season with a focus on subglacial discharge plumes as a mechanism for mediating macronutrient delivery into the photic zone. Entrainment of nutrient-rich seawater within subglacial discharge plumes from marine-terminating glaciers is demonstrated to be the dominant flux of macronutrients associated with discharge from the GrIS. The effect of increasing discharge on summertime marine productivity within regions along the advective pathway of these outflowing plumes is highly dependent on glacier grounding line depth. This depth is critical in determining both the extent to which macronutrients are entrained within the subglacial discharge plume and whether or not the terminal depth of the plume is within the photic zone.

## Results

**Assessing nutrient deficiency.** It is known that summer microbial communities in the high-latitude North Atlantic are limited by $NO_3$ or Fe[27,28]. Fe concentrations in the surface ocean decrease away from the coastline due to the rapid scavenging of terrestrially derived Fe[29,30], and therefore a shift towards Fe-limitation is generally expected in offshore basins. Yet few empirical experiments have tested the bottom-up nutrient control of phytoplankton communities in coastal waters around Greenland and it remains unclear to what extent Fe or $NO_3$ limits summertime marine primary production there. It has, however, been suggested that the timing of phytoplankton blooms in the Labrador Sea, which receives a large fraction of the freshwater flux from Greenland each summer[31], are linked to glacial Fe supply[4,11].

To assess what resource currently limits summer marine productivity around Greenland, we first investigate the potential spatial extent of Fe stress using a compilation of summertime $NO_3$ and dissolved Fe concentrations, shipboard bioassay experiments, and satellite-derived chlorophyll-a fluorescence quantum yields. In general, residual summertime $NO_3$ concentrations indicate Fe-limitation of marine phytoplankton communities, preventing full $NO_3$ drawdown despite high light availability[27,32]. Such summertime residual $NO_3$ is consistently observed around the south and southeast of Greenland in the high-latitude North Atlantic, centered on the Irminger Basin (Fig. 1a). Elsewhere around Greenland, residual $NO_3$ concentrations are not observed throughout summer. Bioassay experiments, which unambiguously test the short-term response of the in-situ phytoplankton community to small increases in different combinations of bioessential nutrients (e.g., Supplementary Fig. 1), have explicitly confirmed that the Irminger Basin is Fe-limited in summer (Fig. 1b)[27,28], while the only available experiment close to the northeast Greenland shelf break found the summertime community to be $NO_3$ limited (Fig. 1b).

Higher spatial resolution is available for summertime Fe* observations (Fig. 1b). Fe* is the excess of dissolved Fe over $NO_3$ for a specified phytoplankton Fe:$NO_3$ requirement. Large positive values of Fe* thereby indicate a relative excess of Fe over $NO_3$ availability, whereas large negative values of Fe* indicate a deficiency of Fe. The spatial pattern of summertime Fe* is consistent with the results of bioassay experiments (Fig. 1b); coastal values are the most positive (Fe replete, N deficient), offshore values are generally close to zero or slightly negative (Fe-N co-deficiency), and the Irminger Basin exhibits a consistently strong negative Fe* signal (Fe deficient, N replete) (Fig. 1b).

One further indication of Fe stress at even higher resolution can be derived from satellite-derived chlorophyll-a fluorescence

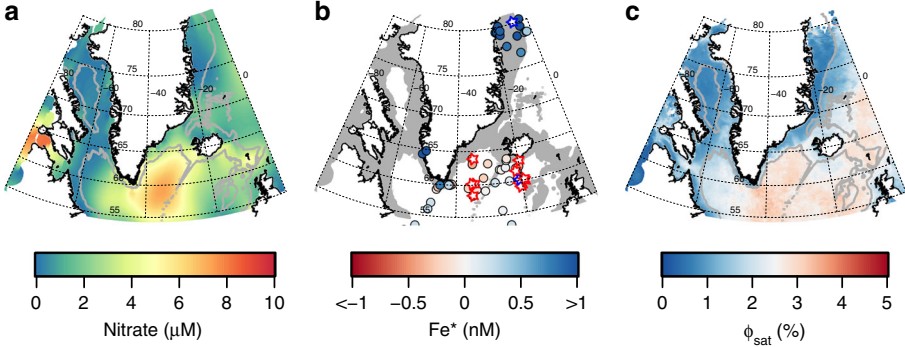

**Fig. 1** Indicators of NO₃ or Fe limitation. **a** Mean World Ocean Atlas (WOA) surface NO₃ concentrations for summer (June, July and August). Gray lines mark the 1 km isobath. **b** Fe* (circles) derived from summertime dissolved Fe and NO₃ data, and observations of Fe limitation (red stars) and NO₃ limitation (blue stars) in shipboard bioassays. Gray shading represents bathymetry <1 km depth. **c** Satellite-derived quantum yield of fluorescence for June, July and August climatological average (2002–2016). Higher quantum yields (red colors) potentially indicate higher Fe stress

| Source | Dissolved nutrients/µM | | | | Ratio N:Si:P:Fe |
|---|---|---|---|---|---|
| | NO₃ | PO₄ | Si | Fe | |
| Arctic river water | 5.7[67] | 0.56[a,b67] | 95[a67] | 0.90 ± 0.85[68,69] | 10:170:1.0:1.6 |
| Ice melt | 1.4 ± 0.9[16] | 0.2 ± 0.2[16] | 13 ± 15[16] | 0.038 ± 0.089[19] | 7.0:65:1.0:0.19 |
| Glacial runoff | 2.0 ± 0.2[3,16] | 0.2 ± 0.1[3,16] | 36 ± 10[a16] | 0.86 ± 1.3[19,70] | 10:180:1.0:4.3 |

**Table 1 Arctic and glacial freshwater composition. Mean (±SD) freshwater nutrient composition in Arctic ice melt, glacial runoff, and river water**

[a]These estimates are discharge volume weighted for respective source catchments
[b]Total dissolved phosphorus (inclusive of PO₄ and dissolved organic phosphorus)

quantum yield (Fig. 1c), which has been demonstrated to correlate with Fe stress of phytoplankton communities in the Southern Ocean[33,34]. A comparison of summertime satellite fluorescence quantum yields around Greenland with residual surface NO₃, shipboard bioassay experiments and Fe* (Fig. 1) indicates a broad-scale matchup, suggesting the technique has promise to qualitatively distinguish between NO₃ and Fe limitation in this system. While fluorescence quantum yield is subject to uncertainties associated with non-photochemical quenching processes, which may vary between ocean biogeochemical provinces, the overall spatial trends offer an additional line of evidence for a summertime pattern of NO₃ limitation in most shelf regions and the deeper enclosed Baffin Bay, which transitions to Fe limitation around the south of Greenland and across the Irminger Basin.

Taken together, the summertime distributions of NO₃ concentrations, Fe*, the results of bioassay experiments, and satellite-derived fluorescence quantum yields support proximal limitation by Fe offshore of the south and southeast Greenland shelf, and by NO₃ in most shelf regions and throughout Baffin Bay (Fig. 1). Therefore, while pre-bloom supply of Fe and potentially other micronutrients to meltwater influenced regions may facilitate summertime bloom initiation[11] and thus enable NO₃ removal, the availability of NO₃ ultimately appears to be the resource constraining integrated summer primary productivity. Accordingly, we next evaluate the significance of different glacier NO₃ supply mechanisms (surface runoff, calved ice and entrainment by subglacial discharge plumes) to the marine environment.

**Glacial runoff composition**. Before estimating meltwater-derived nutrient fluxes, it is prudent to obtain an indication of how important freshwater is as a nutrient source to the high-latitude ocean. The Arctic Ocean constitutes only 1% of the total ocean

volume yet receives 11% of global riverine discharge[35]. Despite disproportionately large river discharge volumes, riverine fluxes of NO₃ and PO₄ to the Arctic are relatively minor[36,37] and are estimated to drive <0.83% of annual basin-wide primary production[36]. GrIS discharge is presently ~1000 km³ per year. Approximating that GrIS discharge occurs as 60% solid ice discharge and 40% runoff[2], this results in a NO₃ flux to the coastal ocean of 1.6 ± 0.6 Gmol (Table 1). For comparison, a flux of 7.0 Gmol NO₃ per year enters the Arctic from its major river systems[36]. Therefore, the NO₃ flux from GrIS discharge itself is very small in terms of the potential effect on large-scale marine primary production.

Normalizing the molar ratio of nutrients to PO₄ (Table 1) demonstrates that glacial runoff is similar to Arctic river water but, in relative terms, enriched in Fe. Relative to the extended Redfield ratio of 16:15:1:0.001 (N:Si:P:Fe, the value for Fe is derived from North Atlantic profiles)[38–40] glacial runoff provides an imbalanced nutrient supply with Fe and Si in excess of NO₃ and PO₄[16]. However, in absolute terms, the concentrations of all nutrients, including Fe and Si, are still relatively low (Table 1), and quite variable (depending on factors including catchment bedrock geology, degree of abrasion, and runoff particle load).

Another effect of glacier systems on the marine environment, linked to the relative enrichment of Si and Fe concentrations, is the formation of turbid sediment plumes that contain high concentrations of fine glacial flour[41]. These plumes are the primary source of elevated Fe in glaciated catchments (Fig. 2a)[42]. However, turbid plumes also suppress penetration of light into surface waters. Therefore, with Fe replete phytoplankton communities in near-shore settings (Fig. 1), an increase in the spatial and temporal extent of turbid plumes could decrease primary production[43].

A general spatial expansion of sediment plumes is expected as surface runoff increases, but clear relationships between runoff

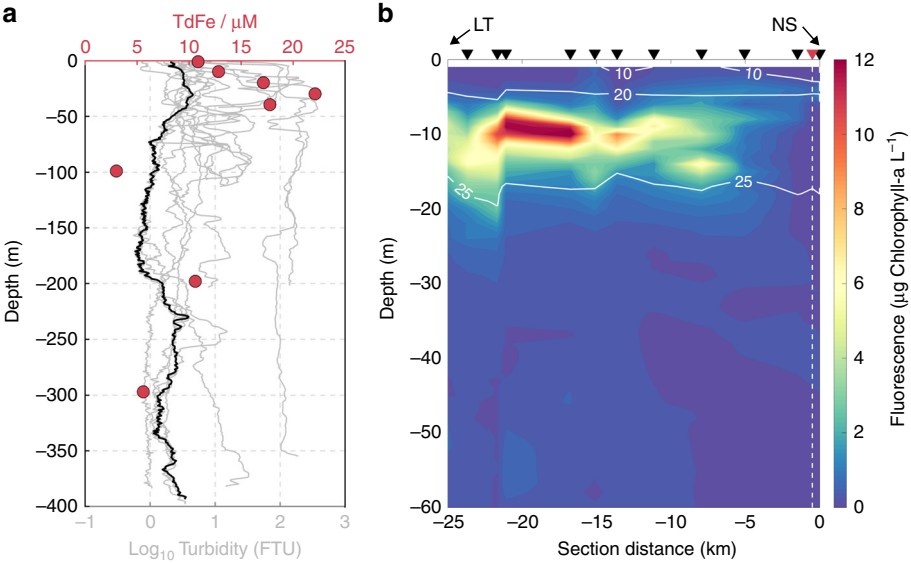

**Fig. 2** Water properties in inner Godthåbsfjord (SW Greenland). **a** Turbidity profiles (gray lines) from all available data within ~5 km of the calving face of the marine-terminating glacier Narsap Sermia (NS), and total dissolvable iron (TdFe, red dots) at a station ~6 km away from NS together with turbidity for this station (black line, June 2015). **b** Fluorescence (June 2015) indicating the presence of a summer phytoplankton bloom. Contours show potential density anomaly (kg m$^{-3}$−1000). NS is located 5 km further into the fjord, Lake Tasersuaq (LT) outflow is also indicated. The dashed line and red triangle correspond to the location of the black turbidity and TdFe data in **a**

volume and plume extent are not always evident for individual catchments[41,44], which complicates prediction of future water column turbidity. Reasons for this may be the limitations of remote sensing techniques, exhaustion of sediment supply through summer and high variability in sediment loads between individual catchments[45,46]. A further contributing factor may be the episodic nature of high turbidity events. A compilation of 16 profiles collected in close proximity (5 km or less from the calving front during 2008–2016) to Narsap Sermia (NS), a marine-terminating glacier in Godthåbsfjord, reveals the highly variable nature of high turbidity in the water column (Fig. 2a). This suggests that a major driver of high turbidity is discrete events, which implies that the correlation between water column turbidity and discharge volume may be highly variable between individual catchments.

In summary, surface runoff and ice melt from glacier systems constitute Fe-rich, NO$_3$-deficient nutrient sources (Table 1)[16] associated with high turbidity. In near-shore systems, where Fe supply from terrestrial sources is already likely sufficient to meet phytoplankton demand, increases in Fe-rich, NO$_3$-deficient discharge would not be expected to drive increases in summer-time productivity. Yet GrIS discharge can also affect nutrient budgets via other mechanisms, such as strengthening stratification and, in the exclusive case of marine-terminating glaciers, the upwelling of nutrients by subglacial discharge plumes.

As subglacial discharge plumes are injected into the water column at the glacier grounding line, they entrain large volumes of subsurface seawater in a buoyant, rising plume. This entrainment can induce substantial upwelling of high NO$_3$ ambient waters near marine-terminating glacier termini[15]. If significant entrainment of deep, nutrient-rich marine waters occurs below the nutricline, and the resulting plume is sufficiently buoyant[8,47,48], this mechanism can constitute the dominant flux of NO$_3$ into a glacier fjord's photic zone and result in sustained phytoplankton blooms throughout summer, as observed in Godthåbsfjord (Fig. 2b)[15,16] and Bowdoin fjord[23]. Critical factors that may affect the magnitude of this process are the distribution of macronutrients in the ambient coastal water column, the depth at which subglacial discharge emerges into coastal waters,

subglacial discharge volume, and the terminal depth (where neutral buoyancy occurs) of the plume[49]. Although extensive water column profiles are available for Godthåbsfjord, there is generally a lack of both biogeochemical and physical data from the immediate vicinity of large marine-terminating glaciers where subglacial discharge plumes first emerge into the water column. Therefore, we next use an idealized plume model to investigate the relationship between subglacial discharge volume and NO$_3$ fluxes.

**Macronutrient fluxes from subglacial discharge plumes.** In order to constrain the upwelling effect of subglacial discharge plumes, we combine summer macronutrient shelf profiles from northeast, southeast, and southwest Greenland (Supplementary Table 1 and Supplementary Fig. 2) with the macronutrient content of glacial ice and runoff (Table 1) in a buoyant plume model for an idealized marine-terminating glacier[49]. It is then possible to estimate the macronutrient flux at the terminal depth of a subglacial discharge plume and the resulting flux into the photic zone (defined as 0–50 m). Four scenarios are evident with respect to how plume upwelling affects downstream macronutrient delivery (Fig. 3). In the first scenario (Fig. 3a), a macronutrient-rich plume is generated, but the glacier is too deep for the plume to remain buoyant and equilibrate in the photic zone. Therefore primary production will not be immediately enhanced by upwelled macronutrients regardless of the magnitude of the upwelling effect.

In the next scenario (Fig. 3b), the glacier is grounded within the optimum zone for enhancing downstream productivity; a large volume of seawater is entrained within the plume, such that a macronutrient-rich plume is generated with sufficient buoyancy that the plume equilibrates in the photic zone. In the third scenario (Fig. 3c), when the grounding line depth has shoaled significantly due to inland glacier retreat, macronutrient upwelling is diminished by two effects. First, the plume entrains a smaller volume of seawater and, second, the entrained waters lie above the nutricline. Finally, when the marine-terminating glacier has retreated inland (Fig. 3d), runoff dilutes the concentration of macronutrients in the surface layer.

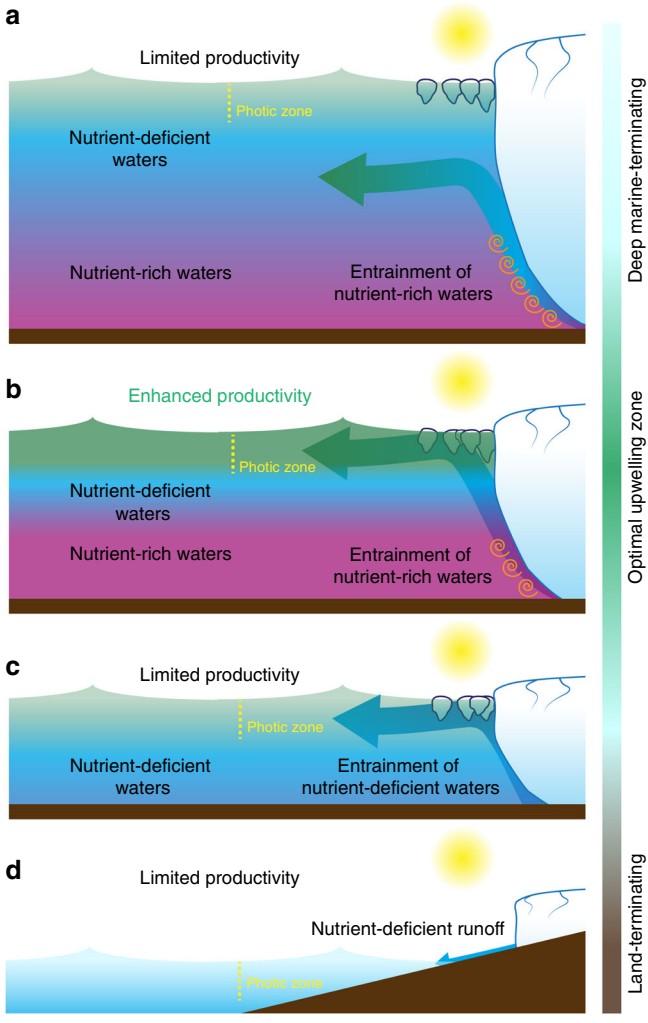

**Fig. 3** Schematic of possible macronutrient upwelling scenarios for idealized marine-terminating glacier systems. **a** Deeply grounded glaciers result in plumes that lose buoyancy through entrainment and equilibrate below the photic zone. **b** Glaciers grounded within the optimum zone for nutrient upwelling facilitate equilibration of a macronutrient-rich plume in the photic zone. **c** Plumes emerging from shallow glaciers equilibrate in the photic zone; however, they are unable to entrain deep, nutrient-rich seawater. **d** Runoff from land-terminating glaciers contain insufficient $NO_3$ to drive enhanced productivity

In order to quantify the relationship between glacier grounding line depth and upwelled macronutrient fluxes, we simulated the plume upwelling effect for an idealized marine-terminating glacier system. In our idealized system, the subglacial discharge rate is fixed at typical summer values and the glacier retreats up an inclined plane such that the rate of inland retreat and decline in grounding line depth are proportional. The plume nutrient flux is calculated as the sum of the macronutrients present in subglacial discharge and the macronutrients entrained within the resulting plume (Fig. 4a). For a modeled $500 \, m^3 \, s^{-1}$ discharge, peak $NO_3$ flux into the photic zone occurs when the glacier grounding line sits at 580 m depth (Fig. 4b). Our results show that a collapse in $NO_3$ supply to the photic zone then occurs if the same discharge enters the ocean at shallower glacier grounding line depths (Fig. 4b). As the glacier retreats inland, and the grounding line shoals from the optimum grounding line depth for $NO_3$ upwelling, large increases in macronutrient supply via subglacial discharge are insufficient to compensate for the

reduced efficiency of the upwelled $NO_3$ supply (Fig. 4a). For a modeled retreat from a 600 to 200 m grounding line depth, neither a twofold nor an extreme tenfold increase in subglacial discharge rate would be sufficient to result in an increased $NO_3$ flux (Supplementary Fig. 3). This is because the glacier shoals to shallow depths with consequently less entrainment of macronutrient-rich seawater by the plume (Fig. 4a). While idealized, these model results demonstrate that plume-driven macronutrient supply to the marine photic zone is not linearly proportional to the subglacial discharge flux.

In almost all cases, even with a weak plume upwelling effect, $NO_3$ and $PO_4$ from freshwater constitutes a small fraction of the net nutrient flux into the photic zone (Fig. 4c). This arises because of the low $NO_3$ and $PO_4$ content of ice melt relative to ambient seawater (Table 1 and Supplementary Fig. 2). Only for shallow marine-terminating glaciers, where the upwelling of macronutrients is ineffective, does the flux of macronutrients from runoff become comparable to the entrained flux (Fig. 4c). This would still be the case even if the macronutrient concentrations in subglacial discharge were enriched several times higher than those used to initialize the model (Table 1). Our model results (Fig. 4) are validated by measured macronutrient concentrations in Godthåbsfjord. For a relatively low entrainment factor (i.e., ratio of upwelled marine ambient water volume to subglacial freshwater volume) of ~14 observed in the proximity of the marine-terminating glacier Narsap Sermia[16], the upwelled flux of nutrients constitutes 87% of the $NO_3$, 95% of the $PO_4$ and 27% of the Si flux into the fjord from all glacial sources (submarine ice melt + surface runoff + subglacial discharge + entrained ambient waters). Similarly for Bowdoin glacier (northwest Greenland), where a smaller entrainment factor of ~6 is observed, the upwelled flux of $NO_3$ constitutes 99% of the $NO_3$ input into the low salinity surface waters close to the marine-terminating glacier termini[23].

As the most critical factor in quantifying upwelled macronutrient fluxes is the entrainment factor, the validity of this approach can also be assessed for Helheim Glacier in Sermilik fjord where an entrainment factor of 30 has been independently determined (August 2015) using noble gases[22]. Using daily estimates of discharge as per Carroll et al.[49], and scaling the entrainment factor to discharge volume, we compute a mean entrainment factor of $30.0 \pm 8.9$ during the peak meltwater season (defined as the time period between which cumulative meltwater discharge rises from 5 to 95% of the annual total). Alternatively, the mean entrainment factor weighted by discharge volume across the year (which includes the early and late melt periods where meltwater input is low $<5 \, m^3 \, s^{-1}$, but the calculated entrainment factors are high >100) is estimated as 34.0. The entrainment factors determined from tracer measurements and plume theory are thus in general agreement.

We next estimate and compare the relative importance of $NO_3$ fluxes from the plume upwelling effect and direct subglacial discharge for 12 glacier systems (Upernavik, Umiamako, Kangiata Nunata Sermia, Kangerdlugssup Sermerssua, Heilprin, Store, Tracy, Helheim, Kangerdlugssuaq, Jakobshavn Isbrae, Alison, and Rink Isbrae) where the grounding line depth and plume properties have been previously characterized over the meltwater season[49]. Here, we combine estimates of subglacial discharge flux with the plume volume at the terminal level, along with conservative estimates of $NO_3$ concentration in the entrained ambient seawater (based on the lower quartile for the mean summer shelf profile at the grounding line depth, Supplementary Fig. 2). The total annual $NO_3$ supply from GrIS runoff and ice melt is approximately 1.6 Gmol, which is often equated to GrIS-to-ocean nutrient "flux"[3,12]. The combined annual subglacial discharge volume from the 12 glaciers for which

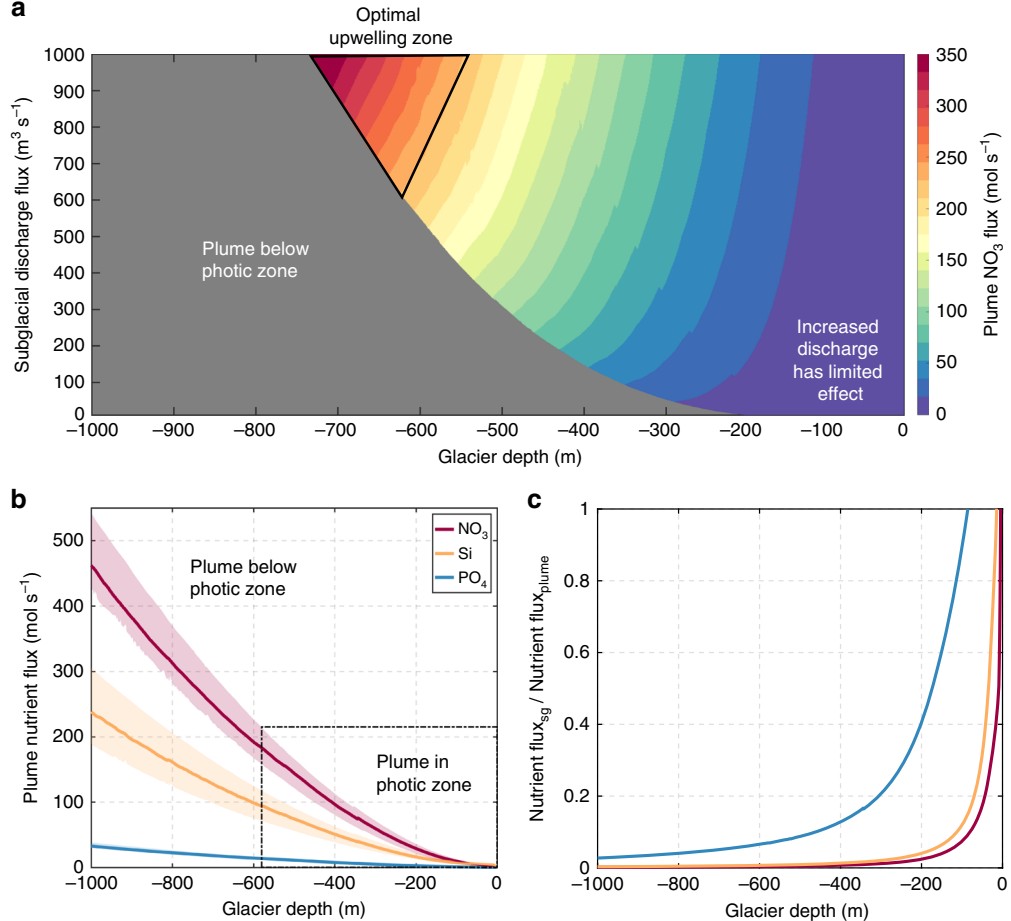

**Fig. 4** Regulation of upwelled macronutrient flux by the subglacial discharge flux and glacier grounding line depth. **a** $NO_3$ flux as a function of subglacial discharge flux ($m^3 s^{-1}$ runoff) and glacier depth (relative to sea-level). The gray region shows where the plume equilibrates below the photic zone (defined as 0–50 m). **b** Macronutrient fluxes for a constant subglacial discharge of 500 $m^3 s^{-1}$, with varying glacier grounding line depth. Upper and lower shaded error bars are defined as 25 and 75% quartiles on macronutrient, temperature, and salinity data (Supplementary Fig. 2). **c** The relative importance of the macronutrient fluxes sourced from subglacial discharge (sg) and from entrainment into the plume. For marine-terminating glaciers, the plume macronutrient flux is generally dominated by nutrients from entrained seawater

plume dilution has been characterized is only 39 km³. Yet upwelling from these 12 glaciers' subglacial discharge plumes produces a combined $NO_3$ flux at the plume terminal depth of ~16 Gmol per year, more than twice the 7 Gmol per year $NO_3$ flux from rivers into the Arctic[36].

The large upwelled nutrient flux from these 12 glaciers also suggests that the prior calculation of Meire et al.[15], who suggested that the total annual upwelled $NO_3$ flux for all marine-terminating glaciers in Greenland was 1.7–17 Gmol, was a vast underestimate. The nutrient entrainment effect was previously underestimated mainly because the entrainment factor of 14 determined for the glacier Narsap Sermia was scaled to the GrIS[15]. Narsap Sermia has a relatively shallow grounding line of only 150–200 m depth[50], so this observed entrainment factor is at the low end of the 4.0–81 discharge-weighted range computed for the 12 systems characterized by Carroll et al.[49] (Supplementary Table 3).

Additional effects, beyond the scope of our conceptual model (Fig. 3), may of course affect the short-term biological response to increased discharge. Turbidity and the depth of the photic zone in particular will change with increasing freshwater input. Nevertheless, our model illustrates the relative importance of the plume upwelling mechanism for supplying macronutrients to coastal regions downstream of the GrIS where summertime Fe supply likely exceeds phytoplankton demand and $NO_3$ is the resource

constraining total summertime primary production (Fig. 1). The upwelled plume $NO_3$ flux from 12 subglacial discharge systems (with total subglacial discharge equivalent to 3.9% of the annual GrIS freshwater discharge volume) provides >1000% of the $NO_3$ flux from freshwater discharge from the entirety of Greenland. Estimated macronutrient fluxes from marine-terminating glaciers based solely on runoff and ice melt nutrient concentrations (Table 1), therefore, vastly underestimate the nutrient flux that can be delivered into the photic zone when marine-terminating glaciers are grounded within the optimum depth range for nutrient upwelling (Fig. 3).

**Changes in grounding line depth and downstream productivity.** Whether future productivity increases or decreases downstream of a marine-terminating glacier under future climate scenarios will depend primarily on whether the present glacier grounding line depth and bed slope are favorable for migration of the grounding line into the optimum zone for upwelling of $NO_3$. Peak $NO_3$ supply will likely already have occurred in glacier systems with shallow grounding line depths above the nutricline (Figs. 3c and 4a). Conversely, peak $NO_3$ supply to the photic zone may have yet to occur in deeply grounded glacier systems, as plumes in these systems may currently upwell large nutrient fluxes, but may not yet reach terminal depths in the photic zone

(Figs. 3a and 4a)[49,51]. Of the 243 distinct Greenland glaciers where bed topography is characterized[50], 148 presently terminate with grounding lines below sea-level with a mean grounding line depth of $280 \pm 200$ m. Of these marine-terminating glaciers, 66 have bed slopes which will remain continuously 200 m below sea-level even with kilometer-scale retreat inland. However, the remaining 55% will transition to land-terminating systems with prolonged retreat[50]. Therefore, for most glaciers, retreat will diminish GrIS-to-ocean nutrient fluxes via the loss of upwelling from subglacial discharge plumes.

Our idealized plume model does not take into account temporal variability in subglacial discharge during marine-terminating glacier retreat[52], modification of shelf waters by fjord-scale processes[51,53], and variability in the subglacial hydrological system, all of which contribute to the uniqueness of each Greenland glacier fjord system. The unique bathymetry and glacier grounding line depth in Greenland's glacier fjords[50] means that the optimum combination of subglacial discharge volume and grounding line depth for maximum $NO_3$ upwelling into the photic zone will likely vary for each individual glacier fjord system. Furthermore, the localized and temporally variable (Fig. 2a) influence of runoff and sediment load on stratification and the depth of the photic zone will add further inter-fjord and temporal variability to upwelled $NO_3$ fluxes[51,54]. Nevertheless, our results demonstrate that upwelling induced by subglacial discharge plumes is a significant $NO_3$ source and will therefore strongly influence $NO_3$-limited summertime primary production.

The spatial scale over which this macronutrient fertilization effect operates will be largely dependent on the extent to which light is limiting productivity in waters close to glacier termini and the flushing time of fjord systems. Both of these factors are spatially and temporally variable around Greenland. Considering the distance over which enhanced macronutrient concentrations can be measured from the few glacier fjords where summertime nutrient distributions have been mapped (e.g., Godthåbsfjord and Bowdoin Fjord)[16,23], we approximate that upwelled nutrient fluxes will potentially enhance primary production over a distance on the order of 10–100 km along the advective pathway of the outflowing plume. This lateral scale will inevitably vary spatially due to the uniqueness of each of Greenland's glacier fjord system's physical features, such as sill depth, bathymetry and fjord length, which exert a strong influence on residence time and fjord-scale circulation[7,50,55].

In conclusion, while it is widely hypothesized that increasing discharge fluxes from the GrIS will fuel elevated marine productivity[2–4,11], the opposite is likely the case for the majority of marine-terminating glacier catchments. Here, we show that summertime GrIS-to-ocean fluxes of $NO_3$, inferred as the primary limiting nutrient for marine productivity around Greenland, are overwhelmingly driven by the entrainment of nutrient-rich marine waters in subglacial discharge plumes rather than by meltwater runoff. $NO_3$ fluxes, therefore, respond non-linearly to changes in GrIS discharge volume.

After accounting for subglacial discharge plume-driven nutrient upwelling, the $NO_3$ flux from 12 major Greenland marine-terminating glaciers is >16 Gmol per year; more than twice the 7 Gmol per year $NO_3$ flux to the Arctic Ocean from Arctic rivers, and an order of magnitude larger than the ~1.6 Gmol per year $NO_3$ contained in discharge from the GrIS. Our results demonstrate that glacier grounding line depth is a strong control on the flux of $NO_3$ from entrainment of ambient seawater in subglacial discharge plumes. Depth exerts a strong influence on both the extent to which macronutrients are entrained within the plume, and on whether the plume achieves neutral buoyancy within the photic zone. A majority of Greenland's marine-terminating glaciers will shoal as they retreat inland from their present termini position in coming decades. Consequently, long-term retreat and shoaling of marine-terminating glaciers is anticipated to diminish a critical source of $NO_3$ to Greenland fjords, leading to reduced summer productivity in fjords and coastal regions affected by subglacial discharge plumes[15].

## Methods

**Satellite data**. All satellite data were MODIS-Aqua level 3 climatologies for summertime (June–July–August) for the 2002–2016 time period, downloaded from the NASA Ocean Color website (https://oceancolor.gsfc.nasa.gov). Fluorescence quantum yields were calculated as described previously[33]. The locations and results of nutrient addition experiments were obtained from prior literature[27,28] and one additional experiment from Fram Strait (Supplementary Fig. 1d).

**Fieldwork and sample analysis**. Shipboard fieldwork was conducted within Godthåbsfjord, southwest Greenland, in June 2015. CTD data was obtained using a Seabird SBE 19plus equipped with a Seapoint turbidity sensor at stations to within approximately 5 km of the marine-terminating glacier Narsap Semia (NS). Profiles from within 5 km of NS were collected opportunistically using the same apparatus in a region confined by 64.65–64.67 °N and 50.05–50.16 °W.

Total dissolvable Fe (TdFe) samples were collected in trace metal clean low density polyethylene bottles (LDPE), acidified by the addition of HCl (UPA, ROMIL) to pH < 2, and stored for 6 months prior to analysis by inductively-coupled plasma mass spectrometry (ICPMS) after dilution with 1 M $HNO_3$ (distilled in-house from SPA, ROMIL)[56]. Dissolved Fe (DFe) samples were collected from Ocean Test Equipment samplers mounted on a plastic coated sampling rosette with a Kevlar conducting cable onboard GEOTRACES section GN05 (new data from GN05 is shown in Supplementary Table 2). Samples were filtered (AcroPak1000 capsule 0.8/0.2 μm filters) and subsequently collected, acidified, and stored in LDPE bottles as per TdFe (above). Analysis via ICPMS was conducted exactly as per Rapp et al.[57] with a combined (buffer + manifold) analytical blank of $64 \pm 18$ pM. Analysis of reference water SAFe produced a Fe concentration of $0.101 \pm 0.016$ nM (consensus value $0.093 \pm 0.008$ nM). In addition to bioassay experiments previously reported for the North Atlantic[27,28], an additional experiment was conducted using the same methodology at 80.2° N, 8.2° W (2–5 August 2016) onboard GEOTRACES section GN05.

**Nutrient data**. Fe* was calculated using Equation 1

$$Fe^* = [DFe] - R_{Fe:N} \times [NO_3^-] \tag{1}$$

A value of 0.069 was used for $R_{Fe:N}$ as this stoichiometry can be derived from North Atlantic water column profiles[40]. DFe and $NO_3$ were obtained from all GEOTRACES compliant summertime data available in the region displayed in Fig. 1[19,58,59]. Macronutrient profiles for shelf waters around Greenland (Supplementary Table 1 and Supplementary Fig. 2) were used to produce a median summer (June–July–August) profile (with upper and lower quartiles) for temperature, salinity, $NO_3$ ($NO_3 + NO_2$), $PO_4$ and Si, which was combined with a subglacial discharge plume model[49].

**Plume model formulation**. We use a steady-state plume model to characterize subglacial discharge plumes rising along a melting, vertical glacier terminus. The governing equations are defined by plume theory, used extensively to describe buoyant plumes in a variety of geophysical settings[60,61]. The model formulation represents a half-conical plume forced by a point source of subglacial discharge (Eqs. 2–4[62]), consistent with observations of discrete subglacial conduits at Greenland glacier termini[63,64]. As the plume rises along the terminus, its volume increases due to the entrainment of seawater and the addition of submarine glacier terminus ice melt. The initial plume temperature and salinity are set to the pressure-salinity-dependent melting point and 0 , respectively; all model parameters are as described previously[62].

To simulate the flux of meltwater into the subglacial discharge plume from glacier terminus melt, we solve a three-equation model[65] describing conservation of heat and salt at the ocean-ice boundary, combined with a liquidus constraint at the interface:

$$\dot{m}(c_i(T_b - T_{ice}) + L) = {}_T C_d^{1/2} c_p u (T_{plume} - T_b), \tag{2}$$

$$\dot{m} S_b = {}_S C_d^{1/2} u (S_{plume} - S_b), \tag{3}$$

$$T_b = \lambda_1 S_b + \lambda_2 + \lambda_3 z, \tag{4}$$

$\dot{m}$ is the melt rate, $u$ is the vertical velocity of the plume, $L$ is the latent heat of fusion, $c_i$ and $c_p$ are the specific heat capacities of ice and water, $T_b$ and $T_{ice}$ are the

ocean-ice boundary and ice temperature, $S_b$ and $S_{plume}$ and are ocean-ice boundary and plume salinity, $C_d^{1/2}\Gamma_T$ and $C_d^{1/2}\Gamma_S$ are the thermal and haline Stanton numbers, $\lambda_{1-3}$, are constants that describe the dependence of freezing point on salinity and pressure, and $z$ is the depth below sea-level.

To characterize macronutrient fluxes, we modify an earlier model formulation[62] by adding three equations (Eqs. 5–7) that represent the change in concentration of $NO_3$, $PO_4$, and $Si$ in the plume:

$$\frac{d}{dz}\left(\frac{\pi}{2}b^2 u NO_3\right) = \pi b \alpha u NO_{3s} + 2b\dot{m} NO_{3m}, \qquad (5)$$

$$\frac{d}{dz}\left(\frac{\pi}{2}b^2 u PO_4\right) = \pi b \alpha u PO_{4s} + 2b\dot{m} PO_{4m}, \qquad (6)$$

$$\frac{d}{dz}\left(\frac{\pi}{2}b^2 u Si\right) = \pi b \alpha u Si_s + 2b\dot{m} Si_m, \qquad (7)$$

$b$ is the plume radius and $\alpha$ is the entrainment constant (set to 0.1). Subscripts of $s$ represent summer nutrient concentrations in seawater (Supplementary Fig. 2) and subscripts of $m$ represent nutrient concentrations in submarine glacier terminus melt (mean ice melt concentrations, Table 1). The first term on the right-hand side represents the entrainment of macronutrients from seawater along the half-plume boundary; the second term represents macronutrient fluxes into the plume from submarine melt along a cross section of the glacier terminus that spans the width of the plume. For all macronutrient equations, the transfer of nutrients through the molecular boundary layer at the ocean-ice interface is assumed to be negligible. The initial plume $NO_3$, $PO_4$, and $Si$ concentrations are defined as mean glacial runoff (Table 1). Finally, we define the terminal depth as the depth where the plume reaches neutral buoyancy and intrudes horizontally into ambient seawater, consistent with results from high-resolution ocean models[48,62,66].

**Data availability**. All new data is available in the main text or the Supplementary Materials.

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

## Acknowledgements

Pablo Lodeiro, Nicola Herzberg, Jaw C. Yong (GEOMAR) Sinhue Torres-Valdes, Kai-Uwe Ludwichowski, and Martin Graeve (AWI) are thanked for nutrient data, and the nutrient enrichment experiment from northeast Greenland. Andrew Yool (Southampton) is thanked for comments on the text.

## Author contributions

Satellite chlorophyll fluorescence quantum yields were processed by T.B., plume upwelling model runs were undertaken by D.C., fieldwork and laboratory analysis was conducted by M.H., L.M., and J.M.. Fe measurements were conducted by S.K. and M.H. M.H., D.C., and E.A. wrote the manuscript with all authors contributing to its revision.

## Additional information

**Competing interests:** The authors declare no competing interests.

