## [Peer Review File · Nature Communications]

Reviewers' comments:

Reviewer #1 (Remarks to the Author):

This manuscript addresses the manner by which rapid changes to the Greenland Ice Sheet will influence marine productivity. I really liked the balance of empirical and modelling work used to underpin the argument, resulting in a clear message and an accessible conceptual model in Figure 3. The paper has the potential to make a significant impact and, in my opinion, a much-needed one at that. This is because there have been several uncritical papers arguing that a big melt flux must mean a big nutrient flux, resulting in marked changes to coastal biogeochemical cycles. Here, then, is a manuscript that demonstrates how the marked changes will in fact result from the physical influence of subglacial water upwelling at tidewater glacier margins (rather than the chemical influence of nutrients being added by the runoff). The fact that glacial meltwater otherwise dilutes the nutrient resource in the fjord is a really important message that I would like to see widely accepted amongst the scientific community. The impact of this paper also comes from pointing out that this circulation-driven fertilization is non-linear, potentially greatly in excess of what we thought was occurring due to runoff inputs, and very sensitive to the "physiography" of the fjord, glacier and coastal environment. The paper gets this message across well, but could be improved with the minor modifications suggested below. I also wonder if the last section (MEDUSA modelling), although insightful, does not stretch the manuscript a little too far. The paper could perhaps end sooner and succinctly on line 268.

- 1) A small point, but the title does not really work for me. The term "depth" needs qualifying (ie depth of what?). I know this needs to be done with 15 words, but can the title better reflect the impact of the paper?
- 2) Line 29. Hodson and others, in Nature Comms 2017, have also pointed out that the delivery of Fe is non-linear using a global data set of riverine glacial Fe (see Figure 3). This is an important point you are making here, so citation of this work would help in my opinion. This builds a more solid foundation to which you add your own original contribution based upon (more important) non-linear fertilisation effects due to deep water entrainment. I would really like to see this non-linearity message come across strongly, so forecasting of future fluxes are properly forced using regional climate model outputs.
- 3) Line 30 should in my opinion say "assumed to be..." so delete "as"
- 4) Line 35. I would add "induced by" before "subglacial drainage". I think this and other amendments to the text will help the reader start to envisage how buoyant plumes initiate a circulation process by dragging up this deep, nutrient-rich water.
- 5) Line 40. You imply that the fertilization effect is due to nutrients from the GrIS with the text "GrIS-to-ocean nutrient flux". Is it not better to explain the greatest uncertainty in the fertilization potential of GrIS-to-ocean WATER (not "nutrient") flux? After all, you are arguing that the NO₃ flux of the glacial inputs is of minimal importance.
- 6) Lines 96 - 97: please re-write the text in parentheses to make your calculations clearer
- 7) Line 99 Use "therefore" instead of "thereby"?
- 8) Figure 3. Is there a case here for a land-terminating glacier, wherein all inputs become riverine? (C.F. the Meire paper). After all, the transition from a tidewater to a land-terminating glacier margin really matters. Also, while I like the simplicity of the figure, is it worth adding the term "nutricline" and indicating shading effects of the turbid plume nearer the glacier margin? Or does this "muddy the waters" too much?

- 9) Line 216: delete "plumes"
- 10) Lines 299 - 303: here I wonder whether you really needed this section
- 11) Line 331: not a sentence.
- 12) Lines 112 and 114: spelling of "phosphorus"

Reviewer #2 (Remarks to the Author):

This paper examines the impact of the Greenland Ice Sheet on the productivity of the surrounding ocean. It is divided into four main sections. In the first, the authors use a combination of remote sensing and secondary shipboard data to argue that in the shallow coastal regions around Greenland NO₃ (rather than Fe) limitation is the primary restriction on productivity. In the second, the nutrient flux of Greenland meltwater is examined; direct runoff from the ice sheet is argued to make a small contribution to Arctic NO₃ compared to regional riverine runoff, an effect further limited by its association with productivity-decreasing sediment plumes. In the third section, a plume model is used to examine the upwelling of nutrients associated with the submarine input of runoff from tidewater glaciers, which is found to be substantially greater than that contained in the runoff itself, and strongly dependent on grounding line depth. In the final section, the authors run a regional ocean biogeochemical model, which is used to discuss potential changes in ocean productivity over coming decades.

Of these, the third section is the one I'm best qualified to comment on. The theory here, and the application of the plume model, is sound. The results are convincing, though not always presented clearly. The finding that the nutrient fluxes in the entrained waters far exceed those in the runoff itself (making the Greenland Ice Sheet much more important as a driver of ocean fertilisation) but may decrease as glaciers retreat seems important, though I will defer to those better versed in this field to judge on the impact and novelty of these conclusions.

I have a number of general comments and questions, some more significant than others:

Title. Why just 'future'?

L11 (and elsewhere). A minor comment on terminology, but I think better to avoid the term 'subglacial plume'. Strictly speaking the plume is proglacial (it's in front of, rather than underneath the glacier). 'Subglacial discharge plumes' (also used in this paper) may be a better term.

L100. The figures given here suggest the GrIS NO₃ flux is equal to ~1/4 of the Arctic riverine NO₃ flux – so certainly smaller, but perhaps not 'negligible'?

L129-134 and Figure 2. I'm not sure that this quite specific case study is a useful contribution to what is otherwise quite a broad scale study. As stated early in the paragraph, it's known that sediment plumes are generally associated with glacial runoff but the relationship between plume extent and runoff can be complex. Adding the extra detail on Godthabsfjord (where the relatively unusual factor of an episodic ice dammed lake drainage may also play a role) doesn't seem to add much and feels out of place.

L182. Where is 'this model result' shown? Should this be a reference to Figure 4a?

L195-206. It hard to match this description with the figures.

-It's not clear where the statement that 'the rate of inland retreat and grounding line depth are assumed to be proportional' applies

-There is no obvious evidence of a tipping point in Figure 4b, or indication of what the nutrient supply from subglacial discharge is

-L201-203 appears to be referencing Figure 4a, but the most recent reference is Figure 4b.

L207. Where are details of these fjord systems given? It would be helpful to include key details in this paper – otherwise it's hard to know if they are shallow/deep or low/high runoff.

L239-243. The phrase 'continuously 200 m below sea level' seems a bit odd – given complete wastage of the ice sheet, all the margins would eventually disconnect from the ocean. And doesn't the significance of these findings (L242-243) stay true for glaciers at which the grounding line becomes shallower but doesn't become land terminating?

L273-303. I'm not convinced that this section adds much. Because the model results presented do not include the processes discussed in the paper, the discussion of implications remains speculative.

L322-326. Why do the authors argue that, despite the vast flux of upwelled NO₃ that they anticipate, the impacts will only be on a local scale? Is the assumption that they upwelled waters will remain local to the glaciers? Studies of fjord circulation (e.g. refs 39, 52) suggest that these plumes flow down-fjord and out to join the shelf circulation. Why would the wider impacts of this be more limited than if the equivalent nutrients were input at the surface?

Figure S3. The upper discharge scenario seems a bit extreme (I don't think there's much evidence for such high meltwater fluxes)

Reviewer #3 (Remarks to the Author):

The manuscript "The depth of Greenland's marine-terminating glaciers regulates future downstream marine productivity" describes an analysis of historical data and some new modeling work to determine the impact of upwelling associated with marine terminating glaciers on nutrient fluxes into the surface ocean during the summer. There are some interesting parts to this paper. I especially liked the section dealing with how existing marine terminating glaciers are likely to evolve in the future to result in less nitrate upwelling and lower productivity.

However, I found the paper confusing and unfocused and I had a hard time figuring out what their main point was supposed to be. In part, I think they were making the point that the depth of the marine terminating glaciers will dictate how much nitrate is upwelled to the surface (more on this below). I think they were also trying to put the upwelling process previously reported in the Meire et al. (2017) paper (upwelling associated with meltwater plumes in marine terminating glaciers has the potential to bring large amounts of nitrate into the surface) into better perspective. A point they made numerous times is that the amount of nitrate upwelled at the front of the glaciers is much higher than the nitrate coming into the Arctic Ocean from rivers or released in glacial meltwater. However, they completely ignore productivity in the spring and the large flux of nitrate that convectively mixes into the surface during the winter that fuels the spring bloom. Given that productivity during the spring bloom is much higher than the productivity during their study period (June-August), the convective nitrate flux should be much larger than the upwelling fluxes they

report here. A back of the envelope calculation of this convective flux, made using observed increases in surface nitrate over the winter (10 μM) and mixed layer depth (20 m) in spring on the Greenland shelf (Harrison and Li 2008), indicate that the 17 Gmol of nitrate reported by the authors to be upwelled by glaciers is equivalent to the amount of nitrate convectively mixed to the surface of the Greenland Shelf over an area of only 85 km². Thus, the convective nitrate flux in winter dwarfs the upwelling nitrate flux in summer. However, as written, the reader is left with the impression that the largest supplier of nitrate for phytoplankton growth in coastal Greenland is upwelling at the front of melting glaciers. The authors should acknowledge that this only applies in summer when all other nitrate fluxes are small.

Even if we restrict our focus to the summer months, the title is misleading in that it states that marine-terminating glaciers regulate downstream production. However, the paper is not at all clear on what is meant by downstream production. Is the glacially-driven nitrate pulse limited to the fjords or does the nitrate escape into the coastal ocean and fuel added production there? The maps shown in Fig 1 and much of the text suggest that their area of interest extends well beyond the fjords, but the text is ambiguous about the fate of the upwelled nitrate. They estimate the upwelling flux at the front of the glacier but never describe the fate of that flux. Does it flow out of the fjord or is it all consumed within?

The authors motivated their study by trying to demonstrate that the coastal waters around southern Greenland are limited by nitrate availability and that iron is limiting in offshore waters to the southeast of the island in the summer. In fact, the first half of the paper was dedicated to this subject. This was done by comparing a map of surface nitrate to a map of the quantum yield of fluorescence, the latter of which is supposed to be a diagnostic of iron stress (it has been shown to be in the Southern Ocean), for the months of June, July, and August. It must be remembered that these months are after the spring phytoplankton bloom that peaks in May around southern Greenland. The authors note that the quantum yield of fluorescence is low nearshore and suggest that this means that iron is not limiting there so nitrate must be the limiting nutrient. The problem with this interpretation is that glacial meltwater enters the coastal zone in June and July, and if it contained appreciable iron, this could explain why there was no sign of iron limitation in their June-August average of the quantum yield of fluorescence. In fact, the image of the quantum yield of fluorescence looks exactly like one would expect if the coastal area around southern Greenland became iron limited after the spring bloom but then this iron limitation was relieved soon after as glacial meltwater rich in iron entered the coastal zone in June and July. Thus, the reason that the coastal zones appear not to be limited by iron could be that the iron stress was already relieved by glacial meltwater. I'm not saying this is correct, but it is another viable interpretation of their data. What is needed is a reliable map (hard to do – see comment later) of the quantum yield of fluorescence in these waters prior to glacial melt but after the spring bloom. Unfortunately, given the narrow window of time between the presumed depletion of nutrients during the spring bloom and glacial melting, this may be difficult to achieve but it is worth a look.

Finally, my impression is that paper is an incremental expansion of the Meire et al. (2017) paper that describes the same upwelling process. The addition of the plume model here was novel, but those results were what one would have predicted in the absence of their model – that only when the bottom of the glacier terminus is below the nitricline is there substantial upwelling of nitrate to the surface.

Specific comments

Line 27-28. Only one source is cited for this statement. If this is not something that is assumed generally by the community, then this statement should be clarified.

Line 58-61. The relationship between satellite-derived quantum yield of fluorescence and iron stress is a complicated one that has not been validated for these waters. The assumption is that phytoplankton fluorescence increases when iron is scarce and absorbed energy cannot freely flow through the photosystems and electron transport chain, which have high iron demands. However,

the measurement of the quantum yield of fluorescence from satellite requires corrections for the fact that other processes also affect how much a given phytoplankton cell will fluoresce (especially under clear skies near the middle of the day when the satellite imagery were collected). These non-photochemical quenching (NPQ) processes will vary depending on the photophysiological state of the phytoplankton. In areas where this technique has been applied successfully (Southern Ocean), numerous measurements of NPQ were used to apply the proper corrections to the satellite-derived fluorescence measurements to obtain quantum yield of fluorescence. As noted by the reference they cite for this method, "If NPQ effects cannot be removed confidently, it is clearly impossible to unambiguously ascribe variability in the quantum yield of fluorescence to other physiological signals such as Fe stress." To my knowledge, these NPQ measurements are not available for their study region and proper corrections have not been made.

Line 87-89. Perhaps, but as noted above, any changes in nitrate delivery referred to here are dwarfed by the amount of nitrate convectively mixed to the surface every year "in the vicinity of Greenland".

Line 100. What is meant by "basin wide"? What basin is being referred to?

Line 93-100. I don't understand why Arctic rivers are included here. Very little of the nutrient runoff from Arctic rivers is likely to directly impact coastal Greenland due to the large distances and long timescales it would take them to get there. I understand it's a number to which glacial upwelling of nitrate can be compared, but it is really not relevant.

Line 107. Si is not vastly in excess of nitrate, at least for diatoms which have a Si:N ratio near 1.

Line 162. What happens to this upwelled nitrate after it reaches the photic zone? Does it move laterally and if so, how far?

Line 204-206. I have a hard time believing that the plume model tells us much that we didn't already know, or at least could have guessed if we gave it a little thought. Did anyone really believe that if the bottom of the glacier was shallower than the nitricline that it could still bring substantial nitrate to the surface? I doubt it.

Line 227-230. True, but this has already been noted by Meire et al. (2017). That paper reports that "Assuming subglacial freshwater discharge is active for approximately two months per year and using a subglacial discharge of 10–100 m³/s per glacier, this results in a nitrate flux of 1.7–17 Gmol/year to the surface layer. Comparing this to the direct input of dissolved inorganic nitrogen with glacial meltwater of 0.3–0.7 Gmol/year (Hawkings et al., 2015), the entrainment associated with subglacial discharge has potentially a far larger impact on the nitrogen supply to the surface layer of Greenland coastal waters." This is precisely the point being made here.

Line 254. What is meant by "near-shore"? Is this outside the fjords or within the fjords?

Line 264. Why the focus on freshwater sources? Other processes are likely to have a much bigger impact on nitrate inventories (changes in upstream production, atmospheric N deposition, etc.). Are these included in MEDUSA?

Line 273. I didn't find this section to be particularly useful. The analysis in the present study applies only to summer months and ignores the most productive period of the year – the spring bloom. I suspect that the MEDUSA results represent differences between annual means, making them very difficult to compare to those presented in this paper.

Line 307-308. Yes, but only after glacial melt water and any trace metals it contained had already entered the coastal zone.

Line 310-314. And tiny when compared to the amount of nitrate convectively mixed to the surface each year on the Greenland shelf.

Reviewer #1 (Remarks to the Author):

This manuscript addresses the manner by which rapid changes to the Greenland Ice Sheet will influence marine productivity. I really liked the balance of empirical and modelling work used to underpin the argument, resulting in a clear message and an accessible conceptual model in Figure 3. The paper has the potential to make a significant impact and, in my opinion, a much-needed one at that. This is because there have been several uncritical papers arguing that a big melt flux must mean a big nutrient flux, resulting in marked changes to coastal biogeochemical cycles. Here, then, is a manuscript that demonstrates how the marked changes will in fact result from the physical influence of subglacial water upwelling at tidewater glacier margins (rather than the chemical influence of nutrients being added by the runoff). The fact that glacial meltwater otherwise dilutes the nutrient resource in the fjord is a really important message that I would like to see widely accepted amongst the scientific community. The impact of this paper also comes from pointing out that this circulation-driven fertilization is non-linear, potentially greatly in excess of what we thought was occurring due to runoff inputs, and very sensitive to the "physiography" of the fjord, glacier and coastal environment.

We thank the Reviewer for their supportive comments. We generally agree that recent literature on the topic has focused on the 'red herring' of nutrient addition from meltwater. The new title and revised conclusions emphasizes our key finding that there is a non-linearity between meltwater input and the total induced nutrient flux to sunlit surface waters. To clearly demonstrate the relative importance of upwelling/entrainment vs the direct nutrient addition from freshwater sources (discharge + ice melt), we add Fig 4C. This clearly showing that runoff is only the dominant nutrient source when the total nutrient flux is small.

The paper gets this message across well, but could be improved with the minor modifications suggested below. I also wonder if the last section (MEDUSA modelling), although insightful, does not stretch the manuscript a little too far. The paper could perhaps end sooner and succinctly on line 268.

The model/end section has been removed. We now refer to global biogeochemical models only to emphasise the impact our new findings will have on the results of their simulations.

1) A small point, but the title does not really work for me. The term "depth" needs qualifying (ie depth of what?). I know this needs to be done with 15 words, but can the title better reflect the impact of the paper?

New title, 'Non-linear response of summertime marine productivity to increased meltwater discharge around Greenland'

2) Line 29. Hodson and others, in Nature Comms 2017, have also pointed out that the delivery of Fe is non-linear using a global data set of riverine glacial Fe (see Figure 3). This is an important point you are making here, so citation of this work would help in my opinion. This builds a more solid foundation to which you add your own original contribution based upon (more important) non-linear fertilisation effects due to deep water entrainment. I would really like to see this non-linearity message come across strongly, so forecasting of future fluxes are properly forced using regional climate model outputs.

Yes, a few lines about Fe and the well-known non-linear behavior of this element across salinity gradients are now included as a prelude to the physical factors leading to non-linearity that we highlight here (new lines 39-56). The revised manuscript now reads: "Fluxes of Fe to the coastal ocean are sustained from both land- and marine-terminating glaciers^{16,17}. It is well demonstrated that significant (~90-99%) losses of this glacially sourced dissolved Fe occur upon mixing with seawater, due to flocculation, which diminishes the flux of this micronutrient¹⁷⁻¹⁹. Less well understood are the physical mixing processes induced by subglacial discharge plumes which may also lead to a complex non-linear relationship between meltwater discharge volume and the magnitude of the induced nutrient fluxes from upwelling²⁰."

3) Line 30 should in my opinion say "assumed to be..." so delete "as"

4) Line 35. I would add "induced by" before "subglacial drainage". I think this and other amendments to the text will help the reader start to envisage how buoyant plumes initiate a circulation process by dragging up this deep, nutrient-rich water.

5) Line 40. You imply that the fertilization effect is due to nutrients from the GrIS with the text "GrIS-to-ocean nutrient flux". Is it not better to explain the greatest uncertainty in the fertilization potential of GrIS-to-ocean WATER (not "nutrient") flux? After all, you are arguing that the NO₃ flux of the glacial inputs is of minimal importance.

- 6) Lines 96 - 97: please re-write the text in parentheses to make your calculations clearer
7) Line 99 Use "therefore" instead of "thereby"?

All minor changes made.

8) Figure 3. Is there a case here for a land-terminating glacier, wherein all inputs become riverine? (C.F. the Meire paper). After all, the transition from a tidewater to a land-terminating glacier margin really matters.

Yes we have done this for clarity (Figure 3 D).

Also, while I like the simplicity of the figure, is it worth adding the term "nutricline" and indicating shading effects of the turbid plume nearer the glacier margin? Or does this "muddy the waters" too much?

We suspect this is too complex because plumes are both surface and sub-surface (with not much knowledge available with respect to how the sub-surface plumes look close to marine-terminating glacier termini) and the nutricline depth will also shift slightly, although we have insufficient data to quantify this. We think a simplistic figure is best to avoid misleading the reader.

10) Lines 299 - 303: here I wonder whether you really needed this section.

As suggested, most of this section is removed.

Reviewer #2 (Remarks to the Author):

This paper examines the impact of the Greenland Ice Sheet on the productivity of the surrounding ocean. It is divided into four main sections. In the first, the authors use a combination of remote sensing and secondary shipboard data to argue that in the shallow coastal regions around Greenland NO₃ (rather than Fe) limitation is the primary restriction on productivity. In the second, the nutrient flux of Greenland meltwater is examined; direct runoff from the ice sheet is argued to make a small contribution to Arctic NO₃ compared to regional riverine runoff, an effect further limited by its association with productivity-decreasing sediment plumes. In the third section, a plume model is used to examine the upwelling of nutrients associated with the submarine input of runoff from tidewater glaciers, which is found to be substantially greater than that contained in the runoff itself, and strongly dependent on grounding line depth. In the final section, the authors run a regional ocean biogeochemical model, which is used to discuss potential changes in ocean productivity over coming decades.

Of these, the third section is the one I'm best qualified to comment on. The theory here, and the application of the plume model, is sound. The results are convincing, though not always presented clearly. The finding that the nutrient fluxes in the entrained waters far exceed those in the runoff itself (making the Greenland Ice Sheet much more important as a driver of ocean fertilisation) but may decrease as glaciers retreat seems important, though I will defer to those better versed in this field to judge on the impact and novelty of these conclusions. I have a number of general comments and questions, some more significant than others:

Title. Why just 'future'? New title, 'Non-linear response of summertime marine productivity to increased meltwater discharge around Greenland'.

L11 (and elsewhere). A minor comment on terminology, but I think better to avoid the term 'subglacial plume'. Strictly speaking the plume is proglacial (it's in front of, rather than underneath the glacier). 'Subglacial discharge plumes' (also used in this paper) may be a better term. 'Subglacial discharge plumes' used throughout.

L100. The figures given here suggest the GrIS NO₃ flux is equal to ~1/4 of the Arctic riverine NO₃ flux – so certainly smaller, but perhaps not 'negligible'? The sentence read 'negligible with respect to Arctic/Atlantic scale productivity, which is correct. The key point was that if the river flux drives 0.8% of Arctic productivity, scaling to meltwater would be 0.2%, but it's actually probably negative because of the NO₃ dilution effect and the fact that surface freshwater plumes reduce the vertical flux of

NO₃, so it really is negligible. In R1 the text now reads “Therefore, the NO₃ flux from GrIS discharge itself is very small in terms of the potential effect on basin-wide primary production.”

L129-134 and Figure 2. *I’m not sure that this quite specific case study is a useful contribution to what is otherwise quite a broad scale study. As stated early in the paragraph, it’s known that sediment plumes are generally associated with glacial runoff but the relationship between plume extent and runoff can be complex. Adding the extra detail on Godthabsfjord (where the relatively unusual factor of an episodic ice dammed lake drainage may also play a role) doesn’t seem to add much and feels out of place. We’re not sure if Godthabsfjord is particularly unusual, it just happens to be well-studied. The figure serves a few uses: demonstrating that a summertime bloom is observed, that the bloom (and Fe) are sub-surface, and that fjord scale waters are very dynamic. With respect to the ice-damm drainage, we know this does occur in Godthabsfjord¹, but we have no indication at all that this was actually responsible for any of the turbidity profiles we illustrate. This paragraph has been re-phased to highlight the uncertainty here.*

L182. *Where is ‘this model result’ shown? Should this be a reference to Figure 4a? Amended, it was previously shown in Fig 4, but we have also explicitly added a new panel to this figure (Fig. 4c) to show that the fractional importance of nutrients in meltwater is low when the upwelling flux is operational.*

L195-206. *It hard to match this description with the figures.*

-It’s not clear where the statement that ‘the rate of inland retreat and grounding line depth are assumed to be proportional’ applies **Rephrased- the concept is that we are constructing a simple model where retreat inland results in steady shoaling of the glacier grounding line (this is not always the case). Now phrased ‘In our idealized system the subglacial discharge rate is fixed at typical summer values and the glacier retreats up an inclined plane such that the rate of inland retreat and decline in grounding line depth are proportional.’**

-There is no obvious evidence of a tipping point in Figure 4b, or indication of what the nutrient supply from subglacial discharge is. Indeed; this has been rephrased as a ‘collapse’.

-L201-203 appears to be referencing Figure 4a, but the most recent reference is Figure 4b. Changed to Fig 4b.

L207. *Where are details of these fjord systems given? It would be helpful to include key details in this paper – otherwise it’s hard to know if they are shallow/deep or low/high runoff. Details now given (see lines 254-) as well as referring to the manuscript that originally presented this².*

L239-243. *The phrase ‘continuously 200 m below sea level’ seems a bit odd – given complete wastage of the ice sheet, all the margins would eventually disconnect from the ocean. And doesn’t the significance of these findings (L242-243) stay true for glaciers at which the grounding line becomes shallower but doesn’t become land terminating? Complete wastage of GrIS is of course not relevant to decadal timescales. The idea we wanted to convey, was that on decadal timescales retreat will lead to shoaling of the grounding line of some glaciers, but not others because of differences in the bathymetry of different glacier beds. We have rephrased this sentence to read ‘have bed slopes which will remain continuously 200 m below sea level even with kilometer-scale retreat inland.’*

L273-303. *I’m not convinced that this section adds much. Because the model results presented do not include the processes discussed in the paper, the discussion of implications remains speculative. We have removed most of this section following collective comments from the three reviewers.*

L322-326. *Why do the authors argue that, despite the vast flux of upwelled NO₃ that they anticipate, the impacts will only be on a local scale? Is the assumption that they upwelled waters will remain local to the glaciers? Studies of fjord circulation (e.g. refs 39, 52) suggest that these plumes flow down-fjord and out to join the shelf circulation. Why would the wider impacts of this be*

more limited than if the equivalent nutrients were input at the surface? It is a vast flux of NO₃ relative to meltwater, but it's still modest compared to the supply from other sources on a large scale (e.g. 'the Arctic' as a whole). There is insufficient data available to show how far these nutrient enriched plumes can be detected (the nutrients get drawdown with time), but existing data suggests a scale of about 10-100 km^{3,4} so we have added a comment about this (new lines 310-318).

Figure S3. The upper discharge scenario seems a bit extreme (I don't think there's much evidence for such high meltwater fluxes) It's meant to be, although we have now added a more realistic two-fold increase as well for clarity. Relative to forecasts of meltwater for 2100 and given the size of glacier termini around Greenland, a tenfold increase in discharge rate would be implausible, but this illustrates the point- that even a massive increase in volume is insufficient to compensate for a loss of glacier depth with respect to the resulting induced nutrient flux.

Reviewer #3 (Remarks to the Author):

The manuscript "The depth of Greenland's marine-terminating glaciers regulates future downstream marine productivity" describes an analysis of historical data and some new modeling work to determine the impact of upwelling associated with marine terminating glaciers on nutrient fluxes into the surface ocean during the summer. There are some interesting parts to this paper. I especially liked the section dealing with how existing marine terminating glaciers are likely to evolve in the future to result in less nitrate upwelling and lower productivity.

However, I found the paper confusing and unfocused and I had a hard time figuring out what their main point was supposed to be. In part, I think they were making the point that the depth of the marine terminating glaciers will dictate how much nitrate is upwelled to the surface (more on this below).

We have now better clarified the main aim of the manuscript: to critically assess how increasing discharge from Greenland will affect summertime marine productivity in terms of nutrient supply. We note that many of the Referees comments below allude to the spring bloom, whereas we considered it logical to focus only on the summertime period, because this is when most meltwater enters the ocean around Greenland and thus when the largest impact of meltwater on marine biogeochemistry might be expected. As the Referee correctly states, a key outcome of our study is that the depth of a glacier grounding line is a key control on how the glacier affects downstream NO₃ availability.

I think they were also trying to put the upwelling process previously reported in the Meire et al. (2017) paper (upwelling associated with meltwater plumes in marine terminating glaciers has the potential to bring large amounts of nitrate into the surface) into better perspective. A point they made numerous times is that the amount of nitrate upwelled at the front of the glaciers is much higher than the nitrate coming into the Arctic Ocean from rivers or released in glacial meltwater. However, they completely ignore productivity in the spring and the large flux of nitrate that convectively mixes into the surface during the winter that fuels the spring bloom.

We agree with the Referee that the impact of future climate change on wintertime nutrient entrainment, for example via an alteration of surface water buoyancy, and its potential modification of spring bloom characteristics is important. However this is largely beyond the aims of our manuscript, which are to evaluate the immediate impact of enhanced meltwater on summertime marine productivity.

Given that productivity during the spring bloom is much higher than the productivity during their study period (June-August), the convective nitrate flux should be much larger than the upwelling fluxes they report here.

This is not the case close to glaciers where this upwelling mechanism operates; the summer bloom period can be as equally productive as the spring bloom period.³

A back of the envelope calculation of this convective flux, made using observed increases in surface nitrate over the winter (10 μM) and mixed layer depth (20 m) in spring on the Greenland shelf (Harrison and Li 2008), indicate that the 17 Gmol of nitrate reported by the authors to be upwelled by glaciers is equivalent to the amount of nitrate convectively mixed to the surface of the Greenland Shelf over an area of only 85 km². Thus, the convective nitrate flux in winter dwarfs the upwelling nitrate flux in

summer. However, as written, the reader is left with the impression that the largest supplier of nitrate for phytoplankton growth in coastal Greenland is upwelling at the front of melting glaciers. The authors should acknowledge that this only applies in summer when all other nitrate fluxes are small.

It was not our intention to convey the message that meltwater is the most important source of nitrate to the Atlantic as a whole, only that the subglacial discharge-upwelling mechanism is the most important *glacially-derived* source of nitrate. To make sure there is no confusion with this regard we have now made sure that we explicitly use the phrase 'summertime' throughout the text.

With respect to the Referee's calculation, the 17 Gmol flux refers to the flux from 12 large glacier systems. Considering that we are quoting a NO₃ input from only 12 glaciers, an equivalency to the convective flux over a shelf area of 85 km² does not seem so small. Turning around the reviewer's comment, each glacier is upwelling the equivalent of convective mixing over 7 km² (85/12) of the shelf and there are 100s of marine terminating glaciers around Greenland! Furthermore, this flux occurs during summer when light is not limiting productivity thus there is potentially an immediate effect on observed primary production. In R1 we clarify the difference between this and the earlier, incorrect, calculation by Meire et al. (lines 254-272).

Even if we restrict our focus to the summer months, the title is misleading in that it states that marine-terminating glaciers regulate downstream production. However, the paper is not at all clear on what is meant by downstream production.

The new title better conveys the key message that there is non-linearity (due to the effect of glacier grounding line depth on upwelling/entrainment) between meltwater discharge and the net effect on nutrient availability resulting from GrIS discharge.

Is the glacially-driven nitrate pulse limited to the fjords or does the nitrate escape into the coastal ocean and fuel added production there? The maps shown in Fig 1 and much of the text suggest that their area of interest extends well beyond the fjords, but the text is ambiguous about the fate of the upwelled nitrate. They estimate the upwelling flux at the front of the glacier but never describe the fate of that flux. Does it flow out of the fjord or is it all consumed within?

We don't have sufficient data to quantify the ultimate fate of these upwelled nutrients and we have now added a few lines about this in the text (see lines 312-316 of the revised manuscript). This will vary regionally depending on where the glaciers are, and the individual characteristics of glacier fjords around Greenland, which have a broad range of flushing times: "The spatial scale over which this nutrient fertilization effect operates will be heavily dependent on the extent to which light is limiting productivity in waters close to glacier termini and the flushing time of glacier fjord systems, both of which are spatially and temporally variable around Greenland. Considering data from the few glacier fjords where summertime nutrient distributions have been mapped (Godthåbsfjord and Bowdoin)⁴, we suggest that upwelled nutrient fluxes will measurably enrich nutrient concentration over a distance of 10-100 km along the path of the advected meltwater."

The authors motivated their study by trying to demonstrate that the coastal waters around southern Greenland are limited by nitrate availability and that iron is limiting in offshore waters to the southeast of the island in the summer. In fact, the first half of the paper was dedicated to this subject. This was done by comparing a map of surface nitrate to a map of the quantum yield of fluorescence [QYF], the latter of which is supposed to be a diagnostic of iron stress (it has been shown to be in the Southern Ocean), for the months of June, July, and August.

We did not use the satellite-derived QYF alone to infer spatial patterns of iron versus nitrate limitation. These data were used as an additional line of evidence alongside the distribution of residual NO₃ (which alone can be interpreted as evidence for another resource limiting phytoplankton productivity and thus NO₃ drawdown) and a synthesis of the results of nutrient addition bioassay experiments. As an additional line of evidence in the revised manuscript we have compiled a dataset of coincidentally available summertime Fe and NO₃ concentration data to calculate Fe*, the residual iron concentration in seawater following full nitrate consumption assuming typical phytoplankton requirements for the elements. These additional data show positive values (Fe in excess) around the Greenland shelf and negative values (Fe deficient) in the offshore Atlantic basin, supporting our original suggestion that nitrate is more likely to be the limiting nutrient around Greenland in areas with a summertime meltwater signal, whilst Fe is more likely to be the limiting nutrient further offshore in the Irminger Basin (see Fig 1).

It must be remembered that these months are after the spring phytoplankton bloom that peaks in May around southern Greenland. The authors note that the quantum yield of fluorescence is low nearshore and suggest that this means that iron is not limiting there so nitrate must be the limiting nutrient. The problem with this interpretation is that glacial meltwater enters the coastal zone in June and July, and if it contained appreciable iron, this could explain why there was no sign of iron limitation in their June-August average of the quantum yield of fluorescence.

As the Referee alludes to, if an area of the ocean is NO₃ limited, it must of course be the case that there are Fe sources to the region that allow for any nitrate to be removed. However, we are not sure how this effects the findings of our manuscript: regardless of where the Fe on the shelf comes from (it's certainly largely terrestrial in origin), NO₃ in surface waters over the shelf are near fully depleted in summer whilst iron is in excess; thus in order to enhance productivity further fixed nitrogen, rather than additional iron, must be supplied.

In fact, the image of the quantum yield of fluorescence looks exactly like one would expect if the coastal area around southern Greenland became iron limited after the spring bloom but then this iron limitation was relieved soon after as glacial meltwater rich in iron entered the coastal zone in June and July. Thus, the reason that the coastal zones appear not to be limited by iron could be that the iron stress was already relieved by glacial meltwater. I'm not saying this is correct, but it is another viable interpretation of their data.

Again here we reiterate that the shelf appears to be NO₃ rather than iron limited, which means there must be moderately strong Fe supply to allow for complete nitrate depletion. As Fe is not the limiting summertime nutrient adding more would not be expected to increase summertime productivity.

What is needed is a reliable map (hard to do – see comment later) of the quantum yield of fluorescence in these waters prior to glacial melt but after the spring bloom. Unfortunately, given the narrow window of time between the presumed depletion of nutrients during the spring bloom and glacial melting, this may be difficult to achieve but it is worth a look.

Looking at the contrast between two time periods during which the physical properties of the water column and irradiance change would be subject to a much greater error than a summertime only investigation (as in our manuscript) because of an increased reliance on an accurate NPQ correction. In any case, even with checks on how NPQ factors change from spring to summer, this calculation would also rely heavily on knowledge of surface meltwater arrival times in areas of the ocean around Greenland. Recent studies have claimed to model [surface] 'meltwater arrival time'⁵⁻⁷, however these model timings are not realistic as they assume all meltwater is released at the surface. In reality the majority (roughly 50-60%) of meltwater from both icebergs and glaciers is released at depth⁸⁻¹⁰. This makes a large difference to the subsequent behavior/fate of the meltwater. A surface release model thus cannot possibly produce a realistic estimate of when/where this meltwater will 'enter the coastal zone' and equally it is generally incorrect to assume that the nutrients associated with meltwater input enter at the surface^{4,11}. To determine when/where meltwater and/or associated nutrients enter the coastal zone would require fjord-scale resolution of bathymetry and the depths at which freshwater is released into the water column in a regional model, which is presently beyond our capabilities.

Defining this time window alone is therefore very challenging and in any case will vary regionally due to differences in the onset of the meltwater season, and the freshwater residence time in different glacier-fjord systems. Some arbitrary definition of the spring bloom termination date based on satellite derived chlorophyll a could be determined for west/south Greenland within a few weeks- but not the East and northern coastlines where the interference from heavy sea-ice cover near the coastline makes this more challenging. Defining a date of meltwater arrival is not possible based on any available 'real' data. Modelled dates are very difficult to verify around Greenland (except for parts of the western coastline where the meltwater signal is very strong) because the meltwater signal is a minor component of the freshwater signal in the coastal boundary currents ('low salinity' primarily originates from sea-ice melt and Euro-Asian river water exported south along the East Greenland coastline). Without some dedicated isotopic studies, (or, as demonstrated recently, with an inert gas⁸) it is therefore difficult to trace meltwater at high resolution.

Finally, my impression is that paper is an incremental expansion of the Meire et al. (2017) paper that describes the same upwelling process. The addition of the plume model here was novel, but those results were what one would have predicted in

the absence of their model – that only when the bottom of the glacier terminus is below the nitricline is there substantial upwelling of nitrate to the surface.

In the absence of the mechanistic model presented here, rather than the conceptual one presented in Meire et al. (2017), it would be difficult to predict the non-linearity between upwelled nutrients and discharge volume, the threshold depth for observing an increase in upwelling, and impossible to accurately calculate the magnitude of this effect (see later comment).

Looking at literature on the subject of nutrient export from glaciers around Greenland, every single paper published in recent years^{5-7,12} has assumed (with the exception of Dr Meire's work), as is indicated in the perspective of Reviewer 1, that nutrient export increases with discharge volume and it is generally assumed that volume and nutrient delivery are proportional. This is obviously incorrect- sometimes nutrient availability doesn't increase at all, and it certainly isn't a simple linear relationship between GfS discharge and nutrient availability. Thus it's perhaps not quite as obvious to everyone as the reviewer states.

In some cases, a failure to recognize even the basic principle that the nutrient concentration in meltwater is low compared to the seawater end-members entering fjords has led to conceptually incorrect conclusions (e.g. Hawkings¹³ - note the Si 'dissolution' mechanism proposed in a recent article in this journal is apparently 70× larger than that normally observed in estuaries.... But the data trend used to deduce this-an increase in [Si] with salinity- is much more simply explained when one considers that the seawater flowing into the fjord at depth is higher in Si than the meltwater flowing out!). The widespread incorrect scaling of nutrient fluxes to freshwater input, and the failure to recognise that upwelling/entrainment rather than meltwater itself is the main nutrient source, we think clearly demonstrates that this work is a valuable addition to the literature on this subject.

Critiquing the most recent Meire¹⁴ work, it fails to recognize that very deep marine terminating glaciers won't drive immediately enhanced productivity. The calculated upwelling entrainment (see later comment) is also a vast under-estimate as this modelling study shows that's entrainment factors can be much larger than the entrainment factor of 14 from a medium-sized marine-terminating glacier (KNS). The approximations in the most recent Meire¹⁴ work yielded an over-simplistic parameterization of both the entrainment factor and discharge- no dataset was used, just ballpark figures for discharge rate and meltwater season length. (We explain this in the R1 text, see new lines 254-272), yielding only a very rough estimate for the whole Icesheet. Here we use real discharge data (daily resolution) from 12 systems along with accurate entrainment factors defined from these systems². Our relationship between nutrient flux is thereby quantitative, whereas in prior work¹⁴ it was largely qualitative based on comparing nutrient biogeochemistry in 2 systems.

Specific comments

*Line 27-28. Only one source is cited for this statement. If this is not something that is assumed generally by the community, then this statement should be clarified. **Added. It is indeed an almost universal assumption.***

Line 58-61. The relationship between satellite-derived quantum yield of fluorescence and iron stress is a complicated one that has not been validated for these waters. The assumption is that phytoplankton fluorescence increases when iron is scarce and absorbed energy cannot freely flow through the photosystems and electron transport chain, which have high iron demands. However, the measurement of the quantum yield of fluorescence from satellite requires corrections for the fact that other processes also affect how much a given phytoplankton cell will fluoresce (especially under clear skies near the middle of the day when the satellite imagery were collected). These non-photochemical quenching (NPQ) processes will vary depending on the photophysiological state of the phytoplankton. In areas where this technique has been applied successfully (Southern Ocean), numerous measurements of NPQ were used to apply the proper corrections to the satellite-derived fluorescence measurements to obtain quantum yield of fluorescence. As noted by the reference they cite for this method, "If NPQ effects cannot be removed confidently, it is clearly impossible to unambiguously ascribe variability in the quantum yield of fluorescence to other physiological signals such as Fe stress." To my knowledge, these NPQ measurements are not available for their study region and proper corrections have not been made.

We have been more careful not to over-interpret these data in the manuscript. We acknowledge the limitations of this technique over such a broad region without extensive checks on NPQ processes, but stress that this is not our single means

for diagnosing nutrient limitation patterns in the region. It is used in combination with residual NO_3 , bioassay experiments and now a synthesis of Fe^* data. We note (a) the good match between these four independent indicators of Fe stress and (b), that the NPQ correction itself does not significantly modify fields of satellite quantum yield i.e., without the correction there would be a stronger contrast between the shelf (low fluorescence yield) and the Irminger Basin (high fluorescence yield); in other words, the contrast does not arise primarily from NPQ processing (please see Fig. S1).

Line 87-89. Perhaps, but as noted above, any changes in nitrate delivery referred to here are dwarfed by the amount of nitrate convectively mixed to the surface every year "in the vicinity of Greenland".

We agree with this, glacier induced changes are small compared to marine NO_3 consumption/supply from other sources. We have stressed the summertime focus of our paper and removed the modelling section of our manuscript.

Line 100. What is meant by "basin wide"? What basin is being referred to? This was meant to convey 'basin-scale', as it doesn't really matter which Basin (Arctic, North Atlantic... an area of the North Atlantic) was being referred to.

Line 93-100. I don't understand why Arctic rivers are included here. Very little of the nutrient runoff from Arctic rivers is likely to directly impact coastal Greenland due to the large distances and long timescales it would take them to get there. I understand it's a number to which glacial upwelling of nitrate can be compared, but it is really not relevant. It's just a relative comparison to show that glacier meltwater is not actually 'rich' in nutrients (much literature in the field implies that it is^{13,15-17}). Given that much more work has been conducted on Arctic river fluxes of nutrients, and their impact off-shore, we do think it is a useful comparison. The point is that rivers are not widely accepted as a major nutrient source to the high latitude ocean, so why is it frequently claimed that glacier-derived nutrients are important when their contribution is even less? The recent literature is heavily biased towards glacial runoff 'fueling' marine productivity without really quantifying the effect, hence we think it is useful to put it into a more oceanographic perspective.

Line 107. Si is not vastly in excess of nitrate, at least for diatoms which have a Si:N ratio near 1. It is a high Si:N ratio compared to anything we are likely to see anywhere in the surface coastal ocean at ambient pH, thus relative to environmentally observed concentrations it is an excess, and thus in this context we think it is correct to state: 'an imbalanced nutrient supply with Fe and Si in excess of NO_3 and PO_4 '

Line 162. What happens to this upwelled nitrate after it reaches the photic zone? Does it move laterally and if so, how far? All available transects suggest it is consumed within about a 10-100 km-scale of the source^{4,11}, but there is not much data on this. We have added a few lines about this in R1. In any case, after the mixing that takes place in subglacial discharge plumes and in some cases also at shallow fjord mouths, it would be difficult to trace the nutrient input itself much further than this because the enrichment would be small.

Line 204-206. I have a hard time believing that the plume model tells us much that we didn't already know, or at least could have guessed if we gave it a little thought. Did anyone really believe that if the bottom of the glacier was shallower than the nitricline that it could still bring substantial nitrate to the surface? I doubt it. Every single recent manuscript on the subject of nutrient biogeochemistry and export from Greenland (with the exception of Dr Meire's work and the recent Kanna et al paper – now cited^{11,14}), as far as we are aware, has assumed that there is linearity between meltwater volume and nutrient export, and that the net effect of meltwater is always a positive influence on marine productivity. Indeed, it is made obvious in this text that this is not the case, but clearly this is not yet widely understood. The comment [words to effect of] 'increasing discharge around Greenland will increase nutrient fluxes' has become very widespread, even in spite of work showing that this is not the case and even though it should be 'obvious' just from salinity/temperature profiles (e.g.⁹) that the glacier-ocean system is much more complicated than that.

Subglacial discharge has been modelled using plume formulations since the 1950s, and yet at no point did it occur to anyone to introduce nutrient equations into this parametrization, so we suspect what the reviewer means is that it is obvious with hindsight!

Line 227-230. True, but this has already been noted by Meire et al. (2017). That paper reports that “Assuming subglacial freshwater discharge is active for approximately two months per year and using a subglacial discharge of 10–100 m³/s per glacier, this results in a nitrate flux of 1.7–17 Gmol/year to the surface layer. Comparing this to the direct input of dissolved inorganic nitrogen with glacial meltwater of 0.3–0.7 Gmol/year (Hawkings et al., 2015), the entrainment associated with subglacial discharge has potentially a far larger impact on the nitrogen supply to the surface layer of Greenland coastal waters.” This is precisely the point being made here.

This earlier calculation wasn't particularly accurate- it scaled up from a glacier with a quite small entrainment factor (14) compared to those observed elsewhere (30-100^{2,8}). We had not noticed that both manuscripts produce a value of 17 Gmol, but this is a coincidence because they actually refer to completely different numbers. We refer only to the flux from 12 glaciers using real salinity/temperature/discharge data. Meire (2017) produced a ballpark figure (which appears now to have been very conservative) and referred to all of Greenland- hence our number is actually much larger than that published in Meire (2017). (The shorter melt season, low estimates of discharge rate and low entrainment factor used by Meire have a multiplying effect, hence their flux for all of Greenland is comparable to our flux for just 12 large glacier systems).

Line 254. What is meant by “near-shore”? Is this outside the fjords or within the fjords? This varies around Greenland because all glacier-fjords are different, it is therefore very difficult to be more specific. However, for clarity, we have added some lines about the spatial scale over which this process is likely to operate at the end of the discussion (new lines 309-316).

Line 264. Why the focus on freshwater sources? Other processes are likely to have a much bigger impact on nitrate inventories (changes in upstream production, atmospheric N deposition, etc.). Are these included in MEDUSA? This section is no longer in the text. Our aim was to investigate how meltwater affects marine nutrient availability, hence our focus is on freshwater derived nutrients.

Line 273. I didn't find this section to be particularly useful. The analysis in the present study applies only to summer months and ignores the most productive period of the year – the spring bloom. I suspect that the MEDUSA results represent differences between annual means, making them very difficult to compare to those presented in this paper. We are not concerned with the spring bloom because it isn't significantly affected by seasonal meltwater input! In any case, this section has been removed.

Line 307-308. Yes, but only after glacial melt water and any trace metals it contained had already entered the coastal zone. This doesn't affect whether or not the surface waters are NO₃ limited. Our data synthesis strongly supports our suggestion that the shelf is NO₃ limited (see earlier major comments) in summer.

Line 310-314. And tiny when compared to the amount of nitrate convectively mixed to the surface each year on the Greenland shelf. Yes, we agree-although not as tiny as the reviewer implied (see above) before we clarified that this flux refers to only 12 glacier systems- yet the main aim of the text is to investigate how freshwater derived nutrient fluxes will change with increasing GrIS discharge and by far this is the largest effect glaciers have on marine nutrient budgets.

References referred to

1. Kjeldsen, K. K. et al. Ice-dammed lake drainage cools and raises surface salinities in a tidewater outlet glacier fjord, west Greenland. *J. Geophys. Res. Surf.* **119**, 1310–1321 (2014).
2. Carroll, D. et al. The impact of glacier geometry on meltwater plume structure and submarine melt in Greenland fjords. *Geophys. Res. Lett.* **43**, 9739–9748 (2016).
3. Juul-Pedersen, T. et al. Seasonal and interannual phytoplankton production in a sub-Arctic tidewater outlet glacier fjord, SW Greenland. *Mar. Ecol. Prog. Ser.* **524**, 27–38 (2015).
4. Meire, L. et al. High export of dissolved silica from the Greenland Ice Sheet. *Geophys. Res. Lett.* **43**, 9173–9182 (2016).
5. Arrigo, K. R. et al. Melting glaciers stimulate large summer phytoplankton blooms in southwest Greenland waters. *Geophys. Res. Lett.* **44**, 6278–6285 (2017).

6. Luo, H. *et al.* Oceanic transport of surface meltwater from the southern Greenland ice sheet. *Nat. Geosci* **9**, 528–532 (2016).
7. Oliver, H. *et al.* Exploring the Potential Impact of Greenland Meltwater on Stratification, Photosynthetically Active Radiation, and Primary Production in the Labrador Sea. *J. Geophys. Res. Ocean.* (2018). doi:10.1002/2018JC013802
8. Beaird, N. L., Straneo, F. & Jenkins, W. Export of strongly diluted Greenland meltwater from a major glacial fjord. *Geophys. Res. Lett.* **43**, (2018).
9. Straneo, F. *et al.* Impact of fjord dynamics and glacial runoff on the circulation near Helheim Glacier. *Nat. Geosci.* **4**, 322 (2011).
10. Enderlin, E. M., Hamilton, G. S., Straneo, F. & Sutherland, D. A. Iceberg meltwater fluxes dominate the freshwater budget in Greenland's iceberg-congested glacial fjords. *Geophys. Res. Lett.* **43**, (2016).
11. Kanna, N. *et al.* Upwelling of macronutrients and dissolved inorganic carbon by a subglacial freshwater driven plume in Bowdoin Fjord, northwestern Greenland. *J. Geophys. Res. Biogeosciences* **123**, (2018).
12. L., B. J. *et al.* Land Ice Freshwater Budget of the Arctic and North Atlantic Oceans: 1. Data, Methods, and Results. *J. Geophys. Res. Ocean.* **0**, (2018).
13. Hawkings, J. R. *et al.* Ice sheets as a missing source of silica to the polar oceans. **8**, 14198 (2017).
14. Meire, L. *et al.* Marine-terminating glaciers sustain high productivity in Greenland fjords. *Glob. Chang. Biol.* **23**, 5344–5357 (2017).
15. Hawkings, J. R. *et al.* Ice sheets as a significant source of highly reactive nanoparticulate iron to the oceans. *Nat. Commun.* **5**, (2014).
16. Hawkings, J. *et al.* The Greenland Ice Sheet as a hot spot of phosphorus weathering and export in the Arctic. *Global Biogeochem. Cycles* **30**, 191–210 (2016).
17. Bhatia, M. P. *et al.* Greenland meltwater as a significant and potentially bioavailable source of iron to the ocean. *Nat. Geosci.* **6**, 274–278 (2013).

Reviewers' comments:

Reviewer #1 (Remarks to the Author):

The paper gives a succinct, clear message about the potential for reduced fertilization as a consequence of the transition from tidewater to land-based glaciation in Greenlandic fjords. I am still of the opinion that this is an important message on account of the great attention being given to land-ocean interaction in the vicinity of this unstable ice sheet. Furthermore, there are many who believe that more fertilization will result because the retreat of glaciers from fjords and onto land will be associated with more runoff and thus greater nutrient export from the ice sheet. This mis-conception is debunked very well in the paper, which at the same time presents some very useful process information and conceptual models that will make it influential and highly cited. The structural changes to the paper in response to the last review have made it much more succinct and I think this will also lead to greater impact.

Minor points

Line 20/21: for fluency, consider inserting comma before "with" on line 20 and insert "being" before "more than" on line 21.

Line 26: Insert hyphen: "meltwater-derived"

Line 55/56: I would better qualify this statement with the insertion of "ensuring" before "nutrient delivery" – because it doesn't imply the direct supply by the plume itself.

Line 71: define "Fe*" before line 81?

Figure 1: There are locations referred to in the text that could be shown: Labrador Sea and Irminger Basin for example.

Line 97: "offer" not "offers"

Line 137: "phosphorus", not "phosphorous"

Line 154: I think "sometimes linked to lake drainage" might be a better expression, or that some re-wording is necessary. This is because the sediment dynamics of glacial inflows are understudied in Greenland, with there being a disproportionate amount of publications linked to outburst floods. Exhaustion, hysteresis and other sediment supply effects mean the relationship between discharge and suspended sediment concentration is nearly always non-linear. Very pronounced flux events are therefore the norm in glacial systems and don't necessarily require an outburst.

Figure 2 could perhaps work better with the inclusion of some boxes or arrows to depict the location of the upper panel on the lower graph. An alternative could be to remove this figure, but I like to see raw data and I fully appreciate that profiling these systems to produce a full empirical characterization is very difficult indeed. Therefore, while I so think other data are probably available to support this section, the data presented are sufficient as a case study and make the reader appreciate the importance of the heuristic application of the model in the section that follows. The transition from the empirical section based around Figure 2 to the modelling therefore works well.

Line 269. Here the modelling is used to carefully suggest that earlier work on one system with a buoyant subglacial plume has underestimated the Greenland Ice Sheet flux quite significantly. This is a more useful statement than is immediately apparent because it alludes to the pitfalls of using a single system and scaling it to the entire Greenland Ice Sheet (which has been undertaken by a number of other studies). However, the sentence is a little awkward and I wonder if it can be

made clearer that other systems are typically deeper than Narsap Sermia?

In the final section the “peak NO₃ supply” concept provides a very strong basis for considering the direction of future change in different Greenland fjords. On line 308 and 309 I still wonder whether it is best to avoid implying that the subglacial plume is the source of NO₃ because the source NO₃ is not in the subglacial plume but is instead entrained in the circulation system that the plume establishes.

Line 323/324: Surface runoff is potentially misleading. You could be referring to fjords with land-based glaciers, or the surface runoff from tidewater glaciers? Maybe “meltwater runoff” is better?

Reviewer #2 (Remarks to the Author):

Thanks to the authors for addressing the comments I made with respect to the initial version of this manuscript - I am satisfied with these responses.

One small question did arise during re-reading: What is the temporal aspect of the data in Figure 2A? In the text, it states that “A compilation of 16 profiles collected in close proximity (5 km or less from the calving front, 2008-2016) to Narsap Sermia (NS), a marine-terminating glacier in Godthåbsfjord, reveals the highly variable nature of high turbidity in the water column (Fig. 2A).” The figure caption however states “Water properties in inner Godthåbsfjord (SW Greenland, June 2015)”. Which timeframe is correct? I assume the latter, but if it's the former, then this would seem to undermine the statement that turbidity in the water column is highly variable (or at least it would be hard to judge this from these data alone).

Reviewer #3 (Remarks to the Author):

The revised version of the paper “The depth of Greenland’s marine-terminating glaciers regulates future downstream marine productivity” has improved significantly since the last time I reviewed it. Most of my comments have been addressed in the revision, but there are still concerns that need to be considered by the authors.

1) The authors do a lot of work to show that coastal Greenland is limited by nitrate in summer, while the Irminger Sea is limited by iron. However, it is important that the authors be more specific in their designation of timing. By summer, do they mean before glacial melt has entered the fjords and coastal Greenland or after? Clearly the upwelling mechanism they propose would be adding nitrate to fjord waters during and subsequent to glacial melt and if these waters were already nitrate limited, one would expect production due to their proposed mechanism to increase. However, what is not clear is when the data they used to assess the degree of nitrate or iron limitation were collected relative to the onset of glacial melt. Is it possible that the diagnoses of nitrate limitation were made after glacial meltwater had already alleviated trace-metal limitation in early summer, driving the system to nitrate limitation when the bioassays and satellite measurements were made? This possibility is quite different from the story they tell here and needs to be directly addressed. For their sequence of events to work, the diagnoses of nitrate limitation need to be made prior to any glacial melt entering the system. Maybe this is the case, here but I couldn't tell from the data presented. I also couldn't figure out where the bioassay data that showed nitrate limitation close to the Greenland coast came from and when the bioassays were performed.

2) I still think that the research needs to be placed in a larger context. They improved this part of the paper significantly already, but there are still too many places where a reader could easily forget that the focus here is in the summer and not the entire year. One thing that would help

would be a short description of the seasonal cycle of production and then an estimate of the fraction of the annual primary production in these systems that happens in the summer, as was done by Juul-Pedersen (2015). Another thing would be to qualify some of the statements made in the paper. For example, in the conclusion, "Here we show that GrIS-to-ocean fluxes of NO₃, inferred as the primary limiting nutrient for marine productivity around Greenland, are overwhelmingly driven by the entrainment of nutrient-rich marine waters in subglacial discharge plumes rather than by surface runoff." This is true only in the summer.

3) Can the predictions of their upwelling model be validated using observational data from Godthåbsfjord? This would provide confidence in the model's predictions.

Other comments

Line 75-76. This is not true. Harrison and Li (2008) and Harrison et al. (2013) show that there is residual nitrate on the Greenland shelf every month of the year, even in summer when mixed layers are shallow. This strongly suggests limitation by something other than nitrate.

Line 117. Are these nutrient fluxes macronutrients, micronutrients or both?

Line 126. The use of the term "basin-wide" is confusing here because the river runoff is entering the Arctic ocean (arguably one basin) while glacial runoff is entering waters around Greenland (not really a basin but also pretty separated from most of the Arctic basin). So where is basin-wide referring to? All waters of the Arctic plus around Greenland? This needs to be clarified.

Line 127-131. This section is overly confusing because at the beginning of the paragraph it states "Fe and silicic acid (Si) concentrations are also lower in glacial sources compared to Arctic river water" but then later in the paragraph it says "glacial runoff is similar to Arctic river water but, in relative terms, enriched in Fe." This sounds contradictory until one figures out that the first statement refers to absolute Fe concentration and the other to Fe concentration normalized to phosphate. It would be helpful to clarify this section.

Line 245-253. Do these calculations refer to summertime only? This should be clarified.

Line 264-266. So what entrainment factor was used for these calculations? Was it 14 or 30-100? The next few lines make it sound like the value of 14 used by Meire et al. was too low compared to other glacier systems, so my guess is that the higher values were used, but I'm not sure. I'm pretty sure that 14 was used in the calculations in lines 245-253. But now I'm not sure why that value was used if it is too low, as stated in this section.

Line 317. Would this distance mean that the impact is mostly in the fjords or also in the coastal ocean?

We thank the Reviewers for their constructive comments that have much improved the manuscript. Please find details of our revisions below and tracked in the accompanying manuscript file.

Reviewers' comments:

Reviewer #1 (Remarks to the Author):

The paper gives a succinct, clear message about the potential for reduced fertilization as a consequence of the transition from tidewater to land-based glaciation in Greenlandic fjords. I am still of the opinion that this is an important message on account of the great attention being given to land-ocean interaction in the vicinity of this unstable ice sheet. Furthermore, there are many who believe that more fertilization will result because the retreat of glaciers from fjords and onto land will be associated with more runoff and thus greater nutrient export from the ice sheet. This mis-conception is debunked very well in the paper, which at the same time presents some very useful process information and conceptual models that will make it influential and highly cited. The structural changes to the paper in response to the last review have made it much more succinct and I think this will also lead to greater impact.

Minor points

Line 20/21: for fluency, consider inserting comma before "with" on line 20 and insert "being" before "more than" on line 21.

Line 26: Insert hyphen: "meltwater-derived"

Line 55/56: I would better qualify this statement with the insertion of "ensuring" before "nutrient delivery" – because it doesn't imply the direct supply by the plume itself.

Line 71: define "Fe" before line 81? (We have rephrased this so that 'Fe*' does not appear before the definition).*

Figure 1: There are locations referred to in the text that could be shown: Labrador Sea and Irminger Basin for example.

Line 97: "offer" not "offers"

Line 137: "phosphorus", not "phosphorous"

All minor points corrected/addressed. We attempted the addition of labels in the Fig, but it then became quite cluttered.

Line 154: I think "sometimes linked to lake drainage" might be a better expression, or that some re-wording is necessary. This is because the sediment dynamics of glacial inflows are under-studied in Greenland, with there being a disproportionate amount of publications linked to outburst floods. Exhaustion, hysteresis and other sediment supply effects mean the relationship between discharge and suspended sediment concentration is nearly always non-linear. Very pronounced flux events are therefore the norm in glacial systems and don't necessarily require an outburst.

We have amended this sentence to (new lines 169-172) 'This suggests that a major driver of high turbidity is discrete events which means that the strength of the relationship between water column turbidity and discharge volume may be highly variable between catchments'.

Figure 2 could perhaps work better with the inclusion of some boxes or arrows to depict the location of the upper panel on the lower graph. An alternative could be to remove this figure, but I like to see raw data and I fully appreciate that profiling these systems to produce a full empirical characterization is very difficult indeed. Therefore, while I so think other data are probably available to support this section, the data presented are sufficient as a case study and make the reader appreciate the importance of the heuristic application of the model in the section that follows. The transition from the empirical section based around Figure 2 to the modelling therefore works well.

Figure 2 is amended for clarity showing the origin of part (b) from (a).

Line 269. Here the modelling is used to carefully suggest that earlier work on one system with a buoyant subglacial plume has underestimated the Greenland Ice Sheet flux quite significantly. This is a more useful statement than is immediately apparent

because it alludes to the pitfalls of using a single system and scaling it to the entire Greenland Ice Sheet (which has been undertaken by a number of other studies). However, the sentence is a little awkward and I wonder if it can be made clearer that other systems are typically deeper than Narsap Sermia?

Yes, we clarify this and raise this issue here (new lines 306-309). The grounding line depth (150-200 m) is added as this is also available, 'Narsap Sermia has a relatively shallow grounding line of only 150-200 m depth'⁵⁴, so the observed entrainment factor is at the low end of the 4.0-81 range calculated for the 12 systems studied by Carroll et al.,⁵³ (Table S3).' In addition, to clarify some confusion raised in Reviewer 3's comments, the entrainment factors for our 12 systems are now shown in Table S3.

In the final section the "peak NO3 supply" concept provides a very strong basis for considering the direction of future change in different Greenland fjords. On line 308 and 309 I still wonder whether it is best to avoid implying that the subglacial plume is the source of NO3 because the source NO3 is not in the subglacial plume but is instead entrained in the circulation system that the plume establishes.

We have tweaked the wording here to add clarity; 'upwelling induced by subglacial discharge plumes is a significant NO₃ source'.

Line 323/324: Surface runoff is potentially misleading. You could be referring to fjords with land-based glaciers, or the surface runoff from tidewater glaciers? Maybe "meltwater runoff" is better?

We have rephrased as 'meltwater runoff'.

Reviewer #2 (Remarks to the Author):

Thanks to the authors for addressing the comments I made with respect to the initial version of this manuscript - I am satisfied with these responses. One small question did arise during re-reading: What is the temporal aspect of the data in Figure 2A? In the text, it states that "A compilation of 16 profiles collected in close proximity (5 km or less from the calving front, 2008-2016) to Narsap Sermia (NS), a marine-terminating glacier in Godthåbsfjord, reveals the highly variable nature of high turbidity in the water column (Fig. 2A)." The figure caption however states "Water properties in inner Godthåbsfjord (SW Greenland, June 2015)". Which timeframe is correct? I assume the latter, but if it's the former, then this would seem to undermine the statement that turbidity in the water column is highly variable (or at least it would be hard to judge this from these data alone).

We have clarified this poor wording as requested by the reviewers. B is a transect from June 2015, whereas 'A' shows one profile from the June transect overlain with all historical data from within a few km of that June 2015 station. The figure is now annotated to more clearly show the link between the two.

Reviewer #3 (Remarks to the Author):

The revised version of the paper "The depth of Greenland's marine-terminating glaciers regulates future downstream marine productivity" has improved significantly since the last time I reviewed it. Most of my comments have been addressed in the revision, but there are still concerns that need to be considered by the authors.

1) The authors do a lot of work to show that coastal Greenland is limited by nitrate in summer, while the Irminger Sea is limited by iron. However, it is important that the authors be more specific in their designation of timing. By summer, do they mean before glacial melt has entered the fjords and coastal Greenland or after?

We have defined summer consistently throughout the text as June-August. The reason for this simplified approach is that the timing of phytoplankton blooms (during spring and summer) varies around Greenland, as does the timing of initial and peak meltwater. Strictly, several of the glacier systems discussed by Carroll et al (i.e. the 12 systems we use to make calculations from) produce some –all-be-it very limited- meltwater every day of the year so it is conceptually difficult to define summer according to whether there is/is not meltwater input. Similarly, any definition of 'peak' meltwater outflow of course varies

with latitude. In addressing comment 2 below, we have added additional text to describe the seasonal cycle and better frame our study in the context of near-shore systems which exhibit pronounced spring and, later, summer blooms.

Clearly the upwelling mechanism they propose would be adding nitrate to fjord waters during and subsequent to glacial melt and if these waters were already nitrate limited, one would expect production due to their proposed mechanism to increase.

This is correct.

However, what is not clear is when the data they used to assess the degree of nitrate or iron limitation were collected relative to the onset of glacial melt.

All of the data is collected after the 'onset' of melt because in S Greenland the onset-regardless of how it is defined- is always earlier than the summer period.

Is it possible that the diagnoses of nitrate limitation were made after glacial meltwater had already alleviated trace-metal limitation in early summer, driving the system to nitrate limitation when the bioassays and satellite measurements were made?

Yes this is possible. If this is the case, the system is 'driven' into NO₃-limitation all summer. i.e. the entire summer bloom period is NO₃-limited, which is exactly what we summarize in the text.

This possibility is quite different from the story they tell here and needs to be directly addressed. For their sequence of events to work, the diagnoses of nitrate limitation need to be made prior to any glacial melt entering the system. Maybe this is the case, here but I couldn't tell from the data presented.

The reviewer is assuming that if glacier-derived Fe played a role in initiating the summer blooms, it would change our conclusions. Yet if this is the case, which is entirely plausible, it has no effect whatsoever on any of our conclusions:

- 1) The regions with strong meltwater signals would still be NO₃ limited throughout summer when the upwelling mechanism occurs.
- 2) Upwelling would still be the dominant glacier-derived NO₃ source and the addition of more NO₃ would be required to drive more productivity around areas which receive a strong meltwater input.

The key point is that the regions with a strong meltwater signal around Greenland are maintained in a high Fe state throughout summer, therefore more NO₃ must be added to drive more productivity in these regions.

It is perfectly plausible that Fe-limitation occurs before/after the summertime period we have defined, yet if this is case, and meltwater plays a central role in 'switching' any system from Fe to NO₃-limitation during the summer period, more meltwater will only act to re-inforce this. i.e. the system will always be driven into NO₃-limitation during the summertime period, and this NO₃ fluxes will remain the most critical with respect to how productivity will change as meltwater fluxes increase. To further clarify the issue in the manuscript, we have added new lines 47-58 and 121-125.

I also couldn't figure out where the bioassay data that showed nitrate limitation close to the Greenland coast came from and when the bioassays were performed.

This was not explained thoroughly in R1, we apologize for this oversight. A new line is added in methods; 'In addition to bioassay experiments previously reported for the North Atlantic (Refs 26 and 27), an additional experiment was conducted using the same methodology at 80.2° N, 8.2° W (3–5 August 2016) onboard GEOTRACES section GN05.' Note that the positioning of this experiment on the map in Figure 1b was slightly incorrect and this has been corrected in this revision.

2) I still think that the research needs to be placed in a larger context. They improved this part of the paper significantly already, but there are still too many places where a reader could easily forget that the focus here is in the summer and not the entire year. One thing that would help would be a short description of the seasonal cycle of production and then an estimate of the fraction of the annual primary production in these systems that happens in the summer, as was done by Juul-Pedersen (2015).

Framing the work within the context of the seasonal cycle is an excellent suggestion and this hopefully clears up some of the discussion we are having with respect to how meltwater may change Fe stress at the end of the spring bloom (new lines 47-57). We stress though the paucity of data on these near-shore systems. Satellite derived data products are difficult to employ to assess chlorophyll-a biomass changes or calculated primary productivity on scales < 1-2 km, which are typical of Greenland's fjords, or in regions with strong surface water sediment and ice discoloration (basically, in any region where our summertime plumes would be strongly evident). Furthermore there are practically no seasonal time series at these sites (note the Juul-Pedersen data referred to in Godthåbsfjord, which we do not dispute is the best available, concerns a site over 100 km away from the closest marine-terminating glacier outflow). We have added new lines 47-57 to the text, but are constrained by the limited availability of literature here.

Another thing would be to qualify some of the statements made in the paper. For example, in the conclusion, "Here we show that GrIS-to-ocean fluxes of NO₃, inferred as the primary limiting nutrient for marine productivity around Greenland, are overwhelmingly driven by the entrainment of nutrient-rich marine waters in subglacial discharge plumes rather than by surface runoff." This is true only in the summer.

This statement is correct at any time of year when there is any GrIS-to-ocean flux. Because entrainment factors are higher for lower discharge (see Figure S5, Carroll et al., 2016), the fractional importance of entrainment by subglacial discharge increases early/late in the meltwater season when the total NO₃ flux (runoff + ice melt + entrainment) is low.

As per Figure 4c, calculated fluxes considering the total quantity of NO₃ show that under all circumstances where the glacier terminates below the surface, the subglacial discharge driven flux exceeds the flux from freshwater input. What this diagram (4c) does not convey is the dilution in NO₃ that would occur in a very shallow system (less than ~50 m or above sea-level). For these shallow glaciers, freshwater stratifies the water column and decreases the NO₃ concentration in the surface mixed layer- thus there is no positive 'flux' of NO₃. This would be especially the case prior to the spring bloom and after the summer bloom.

In summary, there is no scenario in which a positive NO₃ addition to coastal seawater around Greenland from a glacier source is not dominated by subglacial discharge plumes. However, we recognize that this sentence could be miss-interpreted given that the total flux is small or negligible for much of the year, we reword accordingly, 'Here we show that summertime GrIS-to-ocean fluxes of NO₃'.

3) Can the predictions of their upwelling model be validated using observational data from Godthåbsfjord? This would provide confidence in the model's predictions.

With respect to observational nutrient fluxes, no this is unfortunately not yet possible. A key reason for developing the model is that full depth macronutrient profiles close to any marine-terminating glacier, to our knowledge, do not exist- even for Godthåbsfjord. In Godthåbsfjord the best macronutrient data we have comes from 5 km away from a small glacier (Narsap Sermia) and about 20 km away from the major marine-terminating outflow (KNS). In any case Godthåbsfjord would be a particularly challenging case study because there are 3 distinct marine terminating glaciers each with different grounding lines/entrainment factors but overlapping area of influence in terms of the resulting water column features.

The best test using available data which is completely independent of our own approach is to look at Sermilik fjord where entrainment has been estimated from inert gas measurements (new lines 277-287). Beaird et al., report an entrainment factor of 30 in early August 2015 for this fjord using an inert chemical tracer. Taking the data compilation we have used here (from Carroll 2016), we produced an entrainment factor for Sermilik every day of the year using all available salinity/temperature data. Regardless of how we interpret our data, our model predictions are in excellent agreement with Beaird et al.,

For the main discharge period (defining this simply as the days between which 5 and 95% of the annual meltwater total is released) the mean daily entrainment factor produced by our model fit is 30.0 ± 8.9.

Alternatively, taking the sum of daily entrainment factors over the whole year weighted by the fraction of meltwater released on each day produces an integrated value of 34.0.

Other comments

Line 75-76. This is not true. Harrison and Li (2008) and Harrison et al. (2013) show that there is residual nitrate on the Greenland shelf every month of the year, even in summer when mixed layers are shallow. This strongly suggests limitation by something other than nitrate.

It is not quite clear to us what 'is not true' here. We had not noticed this study and thank the reviewer for the reference, yet the region referred to in the two Harrison manuscripts is within the residual NO₃ area shown in Fig 1A. This line (new line 86) reads 'elsewhere around Greenland', referring to regions other than the 'blob' of excess NO₃ which is centered on the Irminger Basin. The box referred to in Harrison et al., which exhibits excess NO₃ is included in the region which we already demonstrate exhibits excess summertime NO₃ (as shown in Fig. 1A).

Perhaps the reviewer is referring to our short hand notation of the area as 'the Irminger Basin'. We have changed the wording in the text to clarify that while this comment (that the Irminger Basin in Fe limited throughout summer) primarily applies to the Irminger Basin (which we previously referred to as off 'SE Greenland'), the region of excess NO₃ is actually larger than the Irminger Basin alone extending further West and East than would be considered to be geographically part of the Irminger Basin.

Line 117. Are these nutrient fluxes macronutrients, micronutrients or both?

We used 'nutrient' to refer to macronutrients and micronutrients together. We have more carefully referred to 'macronutrients' or 'nutrients' in R2.

Line 126. The use of the term "basin-wide" is confusing here because the river runoff is entering the Arctic ocean (arguably one basin) while glacial runoff is entering waters around Greenland (not really a basin but also pretty separated from most of the Arctic basin). So where is basin-wide referring to? All waters of the Arctic plus around Greenland? This needs to be clarified.

We rephrase to 'large scale marine primary production'. In this case it does not really matter what is referred to because of the dilution issue, i.e. irrespective of what area is used, NO₃ addition from meltwater cannot possibly drive enhanced productivity because it is so dilute. In other words, the 'addition of NO₃' from meltwater is not really an addition at all, regardless of what area it refers to.

Line 127-131. This section is overly confusing because at the beginning of the paragraph it states "Fe and silicic acid (Si) concentrations are also lower in glacial sources compared to Arctic river water" but then later in the paragraph it says "glacial runoff is similar to Arctic river water but, in relative terms, enriched in Fe." This sounds contradictory until one figures out that the first statement refers to absolute Fe concentration and the other to Fe concentration normalized to phosphate. It would be helpful to clarify this section.

We have reversed the statement so that it does not read contradictory.

Line 245-253. Do these calculations refer to summertime only? This should be clarified.

We have explicitly stated these refer to the meltwater season. We note the difference between summertime only meltwater discharge and annual meltwater discharge is very small because the vast majority of meltwater is released over a very narrow time period. An annual calculation is practically the same as a summertime only calculation, in fact in most fjord systems it is exactly the same. For Sermilik as an example (Sermilik is picked as an example because it is a fjord system with a trace of melt every day of the year and thus the difference between any defined time periods is maximal), NO₃ fluxes are as follows:

Meltwater season (April-September) 2.1×10^{15} mol NO₃

Annual (Jan-Dec) 2.3×10^{15} mol NO₃

Summer (June/July/August)

1.8×10^{15} mol NO₃

Line 264-266. So what entrainment factor was used for these calculations? Was it 14 or 30-100? The next few lines make it sound like the value of 14 used by Meire et al. was too low compared to other glacier systems, so my guess is that the higher values were used, but I'm not sure. I'm pretty sure that 14 was used in the calculations in lines 245-253. But now I'm not sure why that value was used if it is too low, as stated in this section.

We did not simply repeat the calculation from Meire et al (2017), the value of 14 was not input into this calculation.

Each glacier system has an entrainment factor calculated for each day of the year, so there are several thousand entrainment factors used, not just one ballpark figure. We use the discharge volume/glacier grounding line depth relationship from the work of Carroll et al. (See Supplementary Fig 5a in the Carroll 2016 manuscript for a nice visualization) and daily resolution meltwater volume for each system from all available data/years (dataset also from Carroll 2016). Combining the two, and summing over the entire meltwater season produces the reported flux.

It is clear this was not adequately explained in R1. In order to present the data compilation used from Carroll et al., (2016) simply, we have presented the annual integrated entrainment factors in Table S3.

Line 317. Would this distance mean that the impact is mostly in the fjords or also in the coastal ocean?

As explained in our previous comments, this varies; glacier-fjords around Greenland are 10-200 km long, for some glaciers the plumes are entering the coastal zone immediately without any 'in-fjord' processing, for others (e.g. Godthåbsfjord) the meltwater is released 100-200 km inside fjords. It's simply impossible to generalize hence why we have stressed the uniqueness of each glacier system throughout the text. There are also temporal complexities, the flushing time of fjords varies throughout the meltwater season, but there simply isn't tracer data on this in order to quantify the lateral distance over which these processes operate and how it evolves throughout the meltwater season.

To raise this issue again here, we add the following new lines (357-360) 'This lateral scale will inevitably vary both spatially and temporally due to the uniqueness of each of Greenland's glacier-fjord systems in terms of the physical features, such as sill depth, which exert a strong influence on residence time and watermass circulation'